complexity/mathematical modelling

optimal transport, coupled networks, nonlinearity

**Author for correspondence:**
Amilcare Porporato
e-mail: aporpora@princeton.edu

# A minimalist model for coevolving supply and drainage networks

Shashank Kumar Anand[1], Milad Hooshyar[2],
Jan Martin Nordbotten[4] and Amilcare Porporato[3]

[1]Department of Civil and Environmental Engineering, [2]Princeton Environmental Institute and Princeton Institute for International and Regional Studies, and [3]Princeton Environmental Institute and Department of Civil and Environmental Engineering, Princeton University, Princeton, NJ, USA
[4]Department of Mathematics, University of Bergen, Bergen, Norway

 SKA, 0000-0003-4022-8811; MH, 0000-0002-3667-0482;
JMN, 0000-0003-1455-5704

Numerous complex systems, both natural and artificial, are characterized by the presence of intertwined supply and/or drainage networks. Here, we present a minimalist model of such coevolving networks in a spatially continuous domain, where the obtained networks can be interpreted as a part of either the counter-flowing drainage or co-flowing supply and drainage mechanisms. The model consists of three coupled, nonlinear partial differential equations that describe spatial density patterns of input and output materials by modifying a mediating scalar field, on which supply and drainage networks are carved. In the two-dimensional case, the scalar field can be viewed as the elevation of a hypothetical landscape, of which supply and drainage networks are ridges and valleys, respectively. In the three-dimensional case, the scalar field serves the role of a chemical signal, according to which vascularization of the supply and drainage networks occurs above a critical 'erosion' strength. The steady-state solutions are presented as a function of non-dimensional channelization indices for both materials. The spatial patterns of the emerging networks are classified within the branched and congested extreme regimes, within which the resulting networks are characterized based on the absolute as well as the relative values of two non-dimensional indices.

## 1. Introduction

Many natural and man-made systems consist of materials being conveyed in and/or out of the domain through preferred routes, which result in the evolution of supply and/or drainage

networks. In some biological systems, motile cells regulate their movement based on the affinity or aversion to specific environmental factors (temperature, chemical/biological signal) [1–4]. Two coexisting materials, moving up and down a signal gradient, drive the formation of the competing networks. In other systems, the material is supplied throughout a domain and gets collected once it has been used (and often also transformed), resulting in the formation of co-flowing supply and drainage networks. Examples include the cardiovascular network of blood and nutrients in animals, the supply-chain network of a commodity from the manufacturer to the customer and the related disposal, as well as the aqueduct and waste-flow network in urban water systems [5–11]. In all these systems, the coexisting networks must evolve or be designed in a way that is coordinated, depending on different constraints, such as the configuration of the distribution region, the cost and modes of transportation for supply and drainage materials, etc.

A great deal of research has explored the quantitative laws that explain the structure of networks in different disciplines, but this has been typically done considering either the supply or the drainage network separately [12–16]. In many cases, the general framework for studying such systems has been a static cost optimization problem typical of optimal transport theory [17–20]. As a result, the topology of the underlying supply or drainage network depends on the definition of the cost, including minimum energy dissipation, geometrical constraints, etc. [21–24]. Recently, there have been efforts to interpret this static principle as the result of a dynamic evolution based on partial differential equation (PDE) [25–28].

Less efforts have been devoted to analyse the coevolution of supply and/or drainage transport systems within a continuous domain, which is complicated by the presence of common and individual factors that affect transportation for both materials, including shape and size of the region, parametrization of production/consumption rate, flux velocities, etc. As a step in this direction, this study aims at formulating and analysing a minimalist model that captures the essential interactions between two materials being conveyed in a continuous domain, where the system can be interpreted either as a counter-flowing drainage system or a co-flowing supply and drainage system. The model can be generalized to incorporate multi-species interplay, but for simplicity we limit the discussion to two-species interactions.

The conceptual framework developed here stems from observing the complex ridges and valleys patterns in topographic landscapes, and the related work in the fields of image processing, geomorphology and hydrology related to the duality between the interlocking network of ridges and valleys [29–31]. For its mathematical formulation, we draw inspiration from the landscape evolution models (LEMs), which have been successful in describing the formation of river and stream networks [11,32–35]. Generalizing these models, we develop a simple system consisting of three nonlinear coupled PDEs with an essential parametrization. We introduce a scalar field in continuous spatial domain that mediates two competing mechanisms of either two counter-flowing drainage or two co-flowing supply and drainage problems. We show the influence of rules of production and/or consumption as well as the boundary conditions on the obtained steady-state network patterns.

The paper is structured as follows. In §2, we first present the conceptual framework for two viewpoints of the model. We construct the three-field mathematical model and define non-dimensional indices to describe the relative importance of various factors that alters the characteristics of the coupled networks. We also show that for unitary value of exponents, the proposed model can be re-written as a two-field model. The steady-state closed-form solutions for non-channelized flows in two and three dimensions are derived in §3. In §4, the numerical simulation results for the two-dimensional and three-dimensional cases are presented and the spatial patterns are analysed for different levels of complexity and branching. Conclusions and future research directions are discussed in §5. In appendix A, we discuss the two-field equivalent formulation for the proposed PDE model and the complexity in the boundary conditions that emerges from this model reduction.

# 2. Mathematical model

## 2.1. Conceptual model inspired by the ridge and valley duality

We begin by considering the geometry of a topographic field (figure 1*a*), visualized as a scalar field $h$ in three-dimensional Euclidean space, where the vertical direction points toward gravity. Avoiding maxima (summits) and minima (pits), the curves of particular significance here are the ridges and valleys, which provide a skeleton for the structure of the drainage network [36,37]. With the assumption of negligible

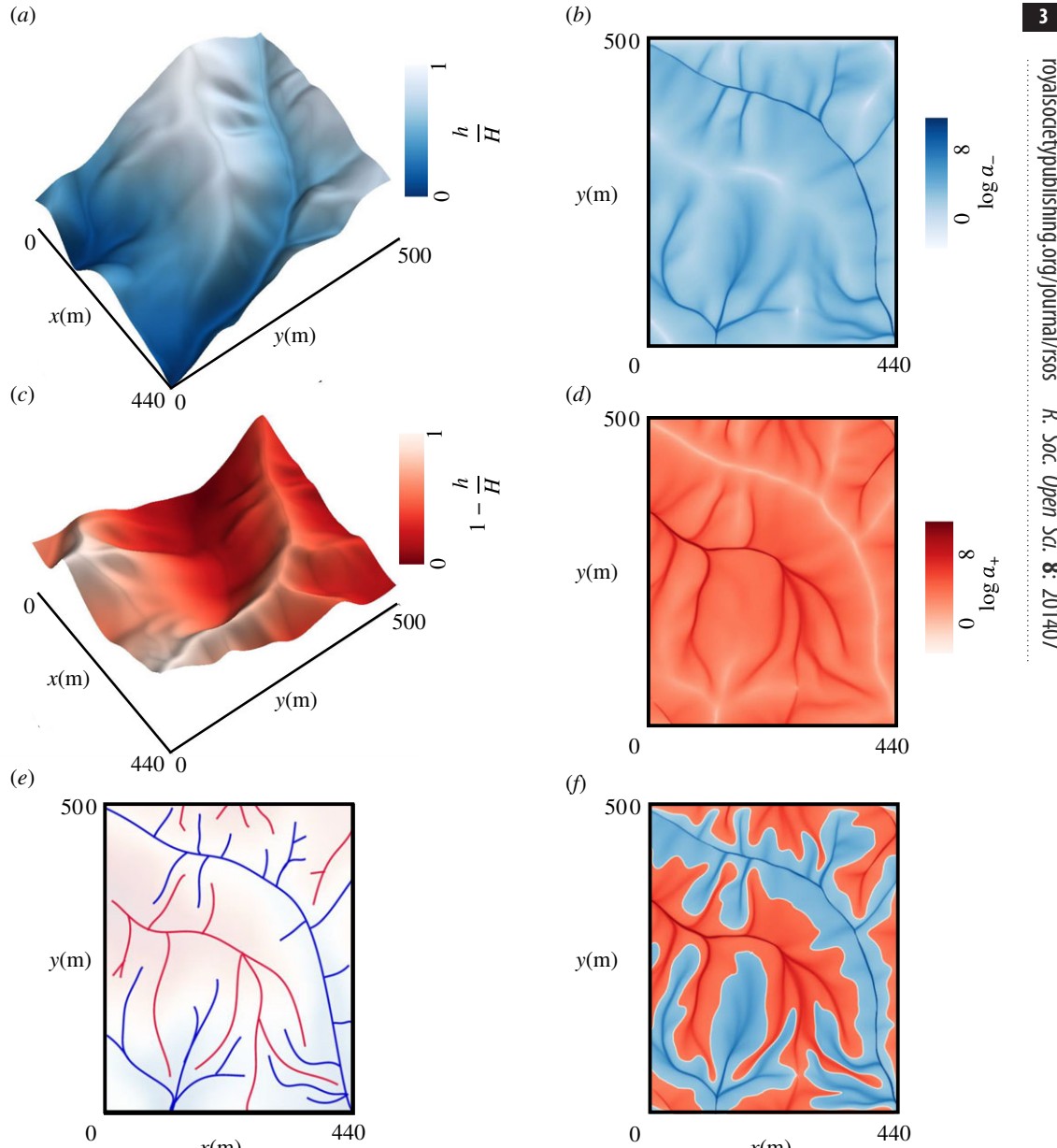

**Figure 1.** Conceptualization of supply and drainage networks using the dual ridge and valley networks in a topographic landscape. (*a*) Three-dimensional surface *h* for the selected topography. (*b*) Drainage network for $a_-$ following the negative gradient of *h*. (*c*) The inverted surface of the original topography, where *H* is the maximum elevation in the domain. (*d*) Drainage network for $a_+$ flowing along the negative gradient of the inverted *h*. (*e*) Interlocked planar ridge and valley networks with prominent ridge-lines (red) and valley-lines (blue). (*f*) The white curve represents the interface $a_+ = a_-$ and separates the red region representing high accumulation of $a_+$ ($a_+ > a_-$) from the blue region showing aggregation of $a_-$ ($a_- > a_+$). The selected topography is from the Calhoun Critical Zone landscape in South Carolina (obtained from the OpenTopography facility (https://opentopography.org/)).

inertial effects, a fluid present in the domain flows under gravity over the scalar field along the direction of the steepest descent. This way, the scalar field, *h*, can be perceived as a potential field guiding the material flow, which results in the distribution of material density, say $a_-$, as shown in figure 1*b*, highlighting the drainage network for the topography. This density, $a_-$, is drained by the stream network and flows out of the system at the boundary.

Inverting artificially the initial topography, as shown in figure 1*c*, the duality between ridges and valleys is apparent, as ridges become valleys and valleys become ridges. The interlocked network of ridge-lines and valley-lines extracted from the original topography is shown in figure 1*e*. Based on this duality, and similarly to the density $a_-$ for the drained material, one can imagine another flow with density $a_+$ in this inverted topography, produced within the domain and drained by the ridge network. The field of material density $a_+$, marking the drainage network for the flipped topography, is

shown in figure 1*d*, where the main courses of flow follow the ridge-lines of the original topography. Therefore, the flow of $a_+/a_-$ moving up/down the slope of the topographic field to be drained by the ridge/valley network is the counter-flow problem (we will refer to this as Problem I).

Reversing the flow direction of the density $a_+$ in the above scenario, the problem can be formulated as a co-flowing supply-drainage problem (Problem II), where $a_+$ represents the density of the supplied material that flows down the slope similar to the drained material of density $a_-$. Instead of having a distributed source through the domain and exiting from the boundary through the ridge network, $a_+$ enters from the boundary where the ridge network forms peaks, and flows along the ridge network following the topographic steepest descent. Figure 1*f* displays the accumulation of $a_+$ along the ridges (red region) and the accumulation of $a_-$ along the valleys (blue region), with the white curve subdividing the regions dominated by either material. One can envision the supplied material density $a_+$ entering the area at the boundary concentrated at the ridges of the scalar field $h$, flowing and getting distributed over the hillslopes as it gets exhausted. In turn, the consumption of the supplied material produces the drained material density $a_-$, which moves under the scalar field potential, and gets discharged out of the domain preferentially via the valleys.

## 2.2. Governing equations

The two-dimensional illustration presented above can be formalized and extended to an $n$-dimensional space ($\mathbf{R}^n$), considering a scalar field $h : \mathbf{R}^n \rightarrow \mathbf{R}$, defined inside a domain $\Omega$ along with two scalar fields $a_+$ and $a_-$ playing the role of the material densities.

For the counter-flow drainage problem (Problem I), the continuity equation for the two materials ($a_+$ and $a_-$), that are produced at a unitary rate and flow with opposite velocity $\mathbf{v}_+$ and $\mathbf{v}_-$, respectively, can be written under the assumption of quasi steady state as

$$\nabla \cdot (a_\pm \mathbf{v}_\pm) = 1. \tag{2.1}$$

For simplicity, we assume that the velocity fields, $\mathbf{v}_+$ and $\mathbf{v}_-$, follow the positive and negative gradient of $h$, respectively, with unit speed as

$$\mathbf{v}_\pm = \pm \frac{\nabla h}{|\nabla h|}. \tag{2.2}$$

The scalar field $h$ is assumed to coevolve with the density fields of both materials. Specifically, the temporal evolution of $h$ consists of a diffusion term and nonlinear sink and source terms due to the feedback from both materials as

$$\frac{\partial h}{\partial t} = D \nabla^2 h + K(r_+ a_+)^{m_+} |\nabla h|^{n_+} - K(r_- a_-)^{m_-} |\nabla h|^{n_-}, \tag{2.3}$$

where $D$ is the diffusion coefficient, $K > 0$, $m_\pm > 0$ and $n_\pm > 0$ are model parameters and $r_\pm$ indicate production rates for the respective material. In this work, we keep these parameters constant over the whole spatial domain. The coupled nonlinear equations (2.1) and (2.3) form a closed system for the interaction of counter-flowing drainage mechanisms by modifying the scalar field $h$ with appropriate initial and boundary conditions for $h$, $a_+$ and $a_-$.

From the viewpoint of the co-flowing supply and drainage mechanism (Problem II), $a_+$ represents the density of the input material in the domain, which is used and drained out of the domain as the output material with density $a_-$. The continuity equation for $a_-$, therefore, remains the same, with the modification in the continuity equation for the input material supplied at the boundaries, that moves with velocity $\mathbf{v}_+$ and gets consumed at the unitary rate, as

$$\nabla \cdot (a_\pm \mathbf{v}_\pm) = \mp 1, \tag{2.4}$$

where both velocity fields, $\mathbf{v}_+$ and $\mathbf{v}_-$, follow the negative gradient of $h$ with unit speed as

$$\mathbf{v}_\pm = -\frac{\nabla h}{|\nabla h|}. \tag{2.5}$$

Equations (2.3) and (2.4) form a general minimalist model for the interaction of two underlying mechanisms of supply and drainage in a spatially continuous domain by modifying the scalar field $h$, as apparent from equation (2.3), where $r_+$ now represents the consumption rate of the supplied material $a_+$. For parsimony, we assume that the supply is consumed uniformly in space at constant

rate, which is immediately disposed giving rise to a uniform and constant source of material that gets drained. More complicated patterns of supply and drainage are certainly of interest and will be investigated in future work. While the model can be analysed from either of two previously discussed formulations, we will mostly consider the viewpoint of supply and drainage mechanisms (Problem II) for the interpretation of the solutions.

A physical understanding of the feedback mechanism related to the source/sink term in equation (2.3) can be achieved by inspecting the example of an eroding overland flow in a natural landscape. The sink term used in landscape evolution models (the same as that employed in equation (2.3)) implies that heavy erosion of $h$ occurs for large values of material density $a_-$ and high magnitudes of $h$ gradients [11,38,39]. Following the steepest descent direction of $h$, more accumulation of the drained material causes high erosion. This feedback loop in carving a preferential path creates a surface instability, which tends to be inhibited by the smoothing effect of diffusion. A threshold exists, above which the instability grows and results in the formation of a complex valley network [11,40].

In this model, the sink and source terms mathematically formalize the same conceptual framework shown in figure 1, where the movement of materials carves out the preferential paths. Thus, for the two-dimensional case, the scalar field $h$ may be viewed as an elevation field of the 'hypothetical' landscape over which input and output materials move following equation (2.5). As indicated by equation (2.3), the accumulation of drainage material decreases the elevation that results in the formation of valleys (sink term). Conversely, the aggregation of the supply material increases the surface elevation that leads to the formation of ridges (source term). Consequently, the input material is accumulated on ridges, while the output material is concentrated in valleys.

In the three-dimensional case, the scalar field can be interpreted as the strength of a chemical signal that drives the movement of the materials (chemotaxis). As equation (2.5) indicates, the concentration of the chemical signal $h$ stimulates the migration of the materials opposite to its gradient. Vascularization of the supply and drainage networks takes place in the domain with high material density of the supply material in high-valued scalar field region and high material density of the drainage material in low-valued scalar field region due to the feedback of sink and source terms in equation (2.3).

The mathematical structure of the proposed model resembles complex models of drainage network evolution, vasculogenesis, chemotaxis and, in general, biological network-formation models as well as surface-growth models [3,41–47]. Specifically, the core component of the model resembles minimalist versions of the well-known Keller–Segel model for chemotaxis under negligible diffusion of biological cells [48–50].

## 2.3. Boundary conditions

The boundary conditions play a crucial role in obtaining solutions for the internal distribution of the densities over the domain. We consider two-dimensional and three-dimensional domains in the shape of a rectangle or parallelepiped, respectively, with the top edge/face ($\Omega_t$) at a fixed higher value ($h = H$) compared with the bottom edge/face ($\Omega_b$) at a fixed lower value ($h = 0$). The remaining side edges/faces ($\Omega_s$) follow zero Neumann boundary conditions in $h$, which provide closed boundary conditions in the densities of the materials ($a_\pm = 0$). For Problem I, the proposed arrangement induces a directionality to the movement of the two materials in the domain with top ($\Omega_t$) and bottom ($\Omega_b$) edges/faces functioning as the exit boundaries for $a_+$ and $a_-$, respectively. Assuming that the densities of the two materials are negligible at their upstream domain boundaries, the boundary conditions become simple and time-independent as $a_+(\Omega_b) = a_-(\Omega_t) = 0$. Under such boundary conditions and the assumption of spatially uniform production rates ($r_+$ and $r_-$), the governing equations compute the counter-flow of the materials across the domain (including at the boundaries where the densities are not specified i.e. $a_+(\Omega_t)$ and $a_-(\Omega_b)$). Figure 2 illustrates the boundary conditions used in this work as well as the corresponding solutions near the first channel instability in the two-dimensional case.

For Problem II, the boundary conditions for $h$ are the same as for Problem I, with top ($\Omega_t$) and bottom ($\Omega_b$) edges/faces functioning now as the entry and exit boundaries for the domain, respectively. Under the assumption that no drainage material exits from $\Omega_t$ and no supply material is conveyed out of $\Omega_b$, the boundary conditions of the material densities remain simple and time-independent as $a_+(\Omega_b) = a_-(\Omega_t) = 0$. The zero Neumann boundary conditions in $h$ on side edges correspond to closed boundary conditions for both material densities ($a_\pm = 0$). Under these conditions, the governing equations determine the flow and distribution of supplied and drained materials over the spatial domain (including at the boundaries where the densities are not specified i.e. $a_+(\Omega_t)$ and $a_-(\Omega_b)$) under the assumption of spatially uniform consumption ($r_+$) and production ($r_-$) rates.

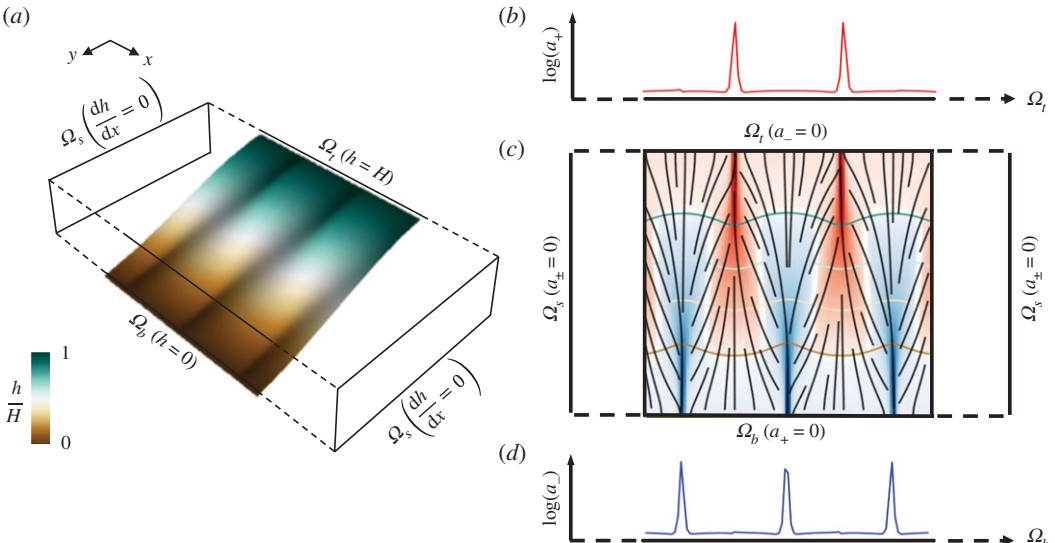

**Figure 2.** Schematic of the boundary conditions used in the model. (*a*) Surface profile of the scalar field $h$ in a portion of the rectangular domain near the first channel instability, where $H$ is the maximum value in the domain (see §4.1 for details). Three shallow ridges and two shallow valleys can be observed in the plotted profile. (*c*) Boundary conditions of $a_+$ and $a_-$ counter-flow drainage problem (Problem I) and co-flow supply and drainage problem (Problem II). Two (red) channels of $a_+$ and three (blue) channels of $a_-$ corresponding to the ridges and valleys in (*a*) are observed. The white curve, representing the interface $a_+ = a_-$, separates the regions dominated by each material. Four contour lines of the scalar field $h$ are plotted along with (black) streamlines which indicate the flow direction of the materials. (*b,d*) Obtained signals of $a_+$ and $a_-$ at $\Omega_t$ and $\Omega_b$, respectively, with peaks indicating channel formation at the corresponding domain boundaries.

It is interesting to observe that the model can be reduced to a two-field system for the case $m_\pm = n_\pm = 1$ and constant $r_\pm$ over the spatial domain. Multiplying the continuity equations for $a_+$ and $a_-$ (equation (2.4)) by $r_+$ and $r_-$, respectively, and subtracting, one can write the single equation for a new spatial field

$$a_* = \frac{r_+ a_+ - r_- a_-}{r_+ + r_-},$$  (2.6)

as

$$-\nabla \cdot \left( a_* \frac{\nabla h}{|\nabla h|} \right) = -1.$$  (2.7)

Equation (2.3) then can be re-written using equations (2.6) and (2.7) as

$$\frac{\partial h}{\partial t} = D\nabla^2 h + K_* a_* |\nabla h|,$$  (2.8)

where $K_* = K(r_+ + r_-)$. Equations (2.7) and (2.8) form a two-field equivalent formulation $(a_*, h)$ to the proposed three-field model $(a_+, a_-, h)$ for unit values of the exponents in equation (2.3). The achieved simplification is, however, only apparent, as the knowledge of $a_+$ and $a_-$ is required in advance to obtain the boundary conditions of the new spatial field $a_*$ in the reduced model corresponding to the solution of the three-field model with the time-independent boundary conditions. We refer to appendix A for a detailed discussion of the boundary conditions for the two-field model in the two-dimensional case.

## 2.4. Non-dimensionalization

For a typical value $H$ of the scalar field and a typical length scale $L$ of the domain, the following dimensionless quantities are established: $\hat{h} = h/H$, $\hat{a}_+ = a_+/L$, $\hat{a}_- = a_-/L$, $\hat{t} = L^2/D$, $\hat{x} = x/L$ and $\hat{y} = y/L$. Using these quantities, equations (2.3) and (2.4) can be written in dimensionless form,

$$\frac{\partial \hat{h}}{\partial \hat{t}} = \hat{\nabla}^2 \hat{h} + \mathcal{C}_{I_+} \hat{a}_+^{m_+} |\hat{\nabla}\hat{h}|^{n_+} - \mathcal{C}_{I_-} \hat{a}_-^{m_-} |\hat{\nabla}\hat{h}|^{n_-},$$  (2.9)

$$-\hat{\nabla} \cdot \left( \hat{a}_\pm \frac{\hat{\nabla}\hat{h}}{|\hat{\nabla}\hat{h}|} \right) = \mp 1,$$  (2.10)

where

$$C_{I_+} = \frac{K r_+^{m_+} L^{2+m_+ - n_+}}{D H^{1-n_+}}, \quad C_{I_-} = \frac{K r_-^{m_-} L^{2+m_- - n_-}}{D H^{1-n_-}}. \tag{2.11}$$

As a result, two 'channelization indices', $C_{I_+}$ and $C_{I_-}$, describe the overall behaviour of the system for the fixed value of the exponents $m_\pm$ and $n_\pm$. As equation (2.9) indicates, an increase in the value of $C_{I_+}$ by high consumption rate, $r_+$, enhances the feedback of the source term. On the other hand, an increase in $C_{I_-}$ by high production rates, $r_-$, strengthens the feedback of the sink term, keeping all other factors the same. This mechanism results in a correlation of the density of the two materials to the value of the scalar field at steady state which can be visualized by looking at the level set $L_c(h)$ of the scalar field $h$ for a constant value $c$. High density of input/output material accruing on the different level sets of the scalar field is shown in the steady-state solutions of the two-dimensional and three-dimensional cases (§4).

# 3. Closed-form solution

At steady state, a closed-form solution can be obtained for the case where diffusion in equation (2.9) inhibits the instability formation in the scalar field. In the two-dimensional case, it can be visualized as the smooth elevation field in a semi-infinite domain where the top edge, which is at a fixed higher elevation $H$ compared with the bottom edge, is separated by the distance $L$ from the bottom edge. This situation is analogous to the flow of two materials before vascularization across two infinite parallel plates placed at a finite distance $L$ in three dimensions, with a fixed high chemical signal strength $H$ at the top face compared with the fixed zero chemical signal strength at the bottom face, which drives the flow of the materials.

Assuming that the scalar field $\hat{h}$ decreases monotonically in the one-dimensional transect, equation (2.10) can be solved with the boundary conditions $\hat{a}_+(\hat{y} = 1) = \hat{a}_-(\hat{y} = 0) = 0$ to obtain $\hat{a}_+ = (1 - \hat{y})$ and $\hat{a}_- = \hat{y}$. For the case of $m_\pm = n_\pm = 1$, substituting the expressions for $\hat{a}_+$ and $\hat{a}_-$ in equation (2.9) at steady state can be written as

$$\hat{h}'' + C_{I_-}\hat{y}\hat{h}' - C_{I_+}(1 - \hat{y})\hat{h}' = 0. \tag{3.1}$$

Solving equation (3.1) gives

$$\hat{h} = \frac{\mathrm{erf}\left(\dfrac{C_{I_-}}{\sqrt{2(C_{I_-} + C_{I_+})}}\right) - \mathrm{erf}\left(\dfrac{C_{I_-}\hat{y} - C_{I_+}(1 - \hat{y})}{\sqrt{2(C_{I_+} + C_{I_-})}}\right)}{\mathrm{erf}\left(\dfrac{C_{I_-}}{\sqrt{2(C_{I_-} + C_{I_+})}}\right) + \mathrm{erf}\left(\dfrac{C_{I_+}}{\sqrt{2(C_{I_-} + C_{I_+})}}\right)}, \tag{3.2}$$

$$|\hat{h}'| = \frac{e^{-\dfrac{(C_{I_-}\hat{y} - C_{I_+}(1 - \hat{y}))^2}{2(C_{I_-} + C_{I_+})}}}{\mathrm{erf}\left(\dfrac{C_{I_-}}{\sqrt{2(C_{I_-} + C_{I_+})}}\right) + \mathrm{erf}\left(\dfrac{C_{I_+}}{\sqrt{2(C_{I_-} + C_{I_+})}}\right)} \times \sqrt{\frac{2(C_{I_-} + C_{I_+})}{\pi}}, \tag{3.3}$$

where $\mathrm{erf}(\cdot)$ is the error function [51].

Smooth profiles using equation (3.2) and the corresponding slope variations following equation (3.3) for $C_{I_+} = 0$, $C_{I_-} = 0$ and $C_{I_+} = C_{I_-} = 25$ are displayed in figure 3a,b. As expected, for $C_{I_-} = 0$ the contribution from the nonlinear sink term goes away and the surface attains a higher profile compared with the case for $C_{I_+} = 0$.

# 4. Numerical solutions

Numerical experiments are started from a linear initial condition containing a small random spatial noise. The added random noise produces small numerical perturbations around the smooth analytical solution (equation (3.2)) so that, when the latter is unstable, channel instabilities grow resulting in the formation of spatial patterns. The boundary conditions in all the numerical simulations for two-/three-dimensional cases are taken the same as discussed in §2.3. Raster grid with a spacing of one unit is used in the numerical simulations for two-/three-dimensional cases.

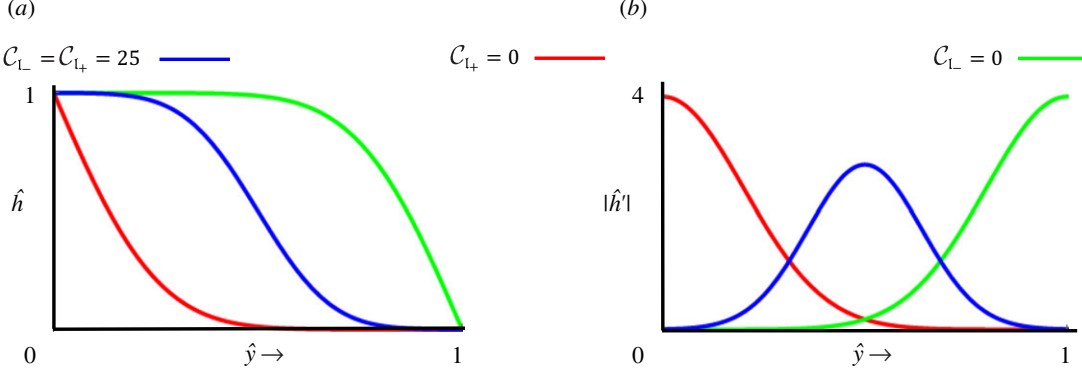

**Figure 3.** (a,b) Steady-state solutions given by equations (3.2) and (3.3) for three cases of $\mathcal{C}_{I_+} = 0$ (red), $\mathcal{C}_{I_-} = 0$ (green) and $\mathcal{C}_{I_\pm} = 25$ (blue).

We use the efficient numerical algorithm presented in [40] to update the scalar field $h$ over the entire domain until the steady state is reached. The numerical scheme consists of a time-splitting approach where we use a five-point stencil second-order central-difference formula for discretizing the Laplace operator for the diffusion and a modified breadth-first topological sorting algorithm for implicitly updating the nonlinear source/sink term. The sorting algorithm presented in [40] belongs to the so-called 'task-scheduling' problems, where the edges of the network symbolize the tasks' dependency. In this model, edges represent the relationship among points in the flow-distribution network of the material, which is traversed in a way to make the matrix system upper/lower triangular for the efficient implicit computation. The development and the applications of this category of sorting algorithm are discussed in [52–56].

The accuracy of the employed numerical algorithm has been carefully tested for the case of the drainage-network evolution model in the natural landscape against analytical solutions in non-channelized/vascularized conditions as well as against analytical results of linear stability analysis [11,40]. The temporal evolution of the mean-field solution in the fully channelized/vascularized regime agrees with the exact analytical expression for the transient solution [40]. We refer to these references for further details.

## 4.1. Two-dimensional case

### 4.1.1. Code verification

We first simulate a rectangular domain with high aspect ratio (length = 500, width = 100) for unitary exponents ($m_\pm = n_\pm = 1$). The mean elevation profile along the length is compared with equation (3.2) for varying values of $\mathcal{C}_{I_\pm}$ to verify the implemented code. The closed-form solution is applicable for the cases when the field $h$ is smooth enough (no channelization). For this case, the first channelization in the domain occurs at the value of $\mathcal{C}_{I_\pm} = 3.5$. The mean surface profile starts deviating from the closed-form solution for $\mathcal{C}_{I_\pm} \geq 3.5$ (figure 4a) due to channel formation. This is apparent from the accumulation plot of $a_+$ and $a_-$, where the white curve represents the interface for $a_+ = a_-$. For $\mathcal{C}_{I_\pm} = 1$, the interface is a straight line, while for $\mathcal{C}_{I_\pm} = 3.5$ (the onset of first channelization) and 12.5, the interface becomes a curve due to the emergence of channels in supply and drainage networks (figure 4b–d).

The coupled supply and drainage networks obtained as steady-state solutions near the first channelization have a non-uniform gap between them, as shown in figure 4. These symmetry-breaking irregularities, or dislocation defects, in the channel spacing are essentially created by the mismatch between the solution geometry with the domain geometry used in the model, a behaviour which is not unexpected in nonlinear pattern-forming systems [57,58]; see [11] for a discussion of the effect of domain shape on the non-symmetric spacing in the drainage network near the first channelization.

We also verify that the spatial randomness added to the linear initial condition to trigger the channel instabilities does not alter the overall solution creating asymmetric valley spacing. Adding random samples from a uniform distribution over $[0, \hat{u}_r]$ for three levels of $\hat{u}_r$ ($10^{-2}$, $10^{-3}$, $10^{-4}$), we observe the steady-state solutions for the same rectangular domain and fixed value of $\mathcal{C}_{I_\pm} = 10$ (near the first

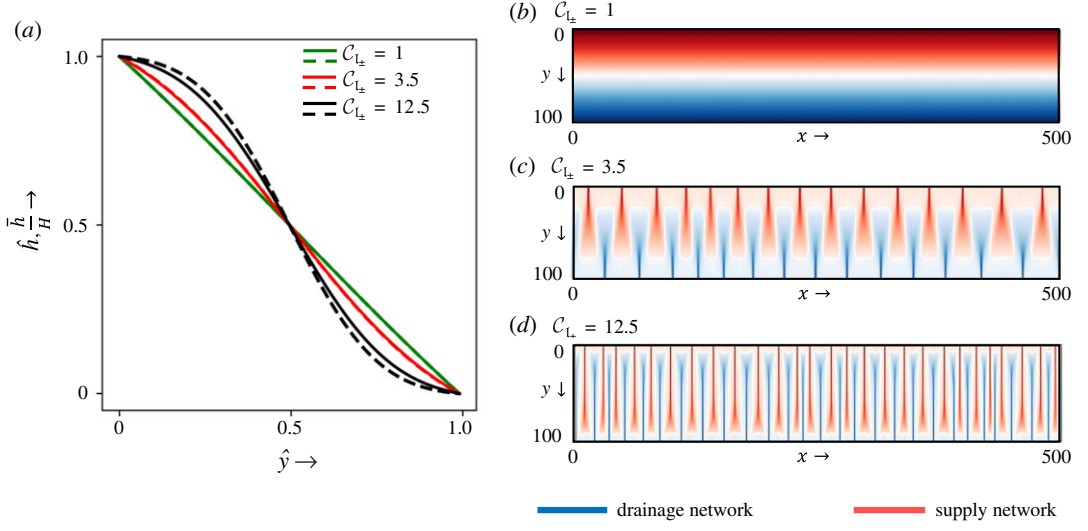

**Figure 4.** (a) Solid lines represent the computed mean surface profile ($\bar{h}$) along the length in a rectangular domain (width = 100, length = 500) with grid spacing $\Delta x = \Delta y = 1$ for $\mathcal{C}_{I_{\pm}} = 1$, 3.5 and 12.5 compared with the steady-state closed-form solutions given by equation (3.2) as dashed lines. (b–d) Simulation results for the accumulation of (red) $a_{+}$ and (blue) $a_{-}$ for $\mathcal{C}_{I_{\pm}} = 1$, 3.5 and 12.5. The white curve represents the interface $a_{+} = a_{-}$, separating the regions dominated by either material.

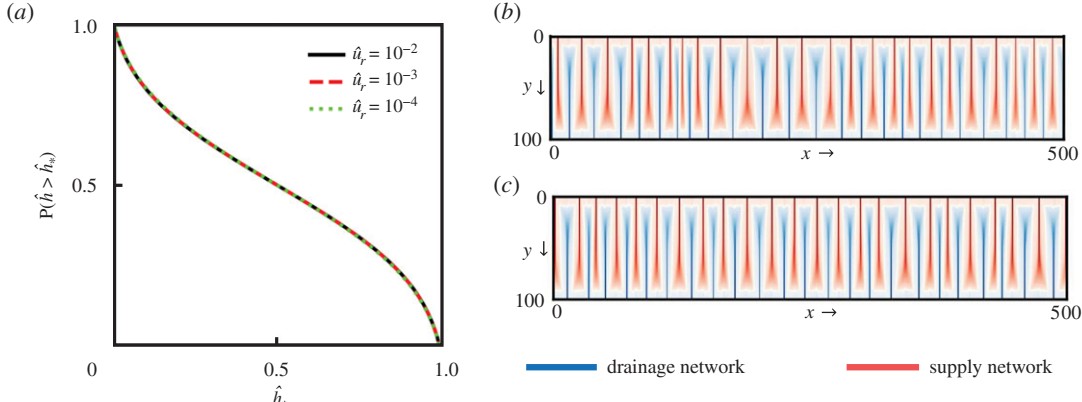

**Figure 5.** Steady-state simulation results for $\mathcal{C}_{I_{\pm}} = 10$ using three different perturbations in the initial conditions for a rectangular domain (width = 100, length = 500) with grid spacing $\Delta x = \Delta y = 1$. (a) Distribution of the scalar field, $h$, for $\hat{u}_{r} = 10^{-2}$ (black), $\hat{u}_{r} = 10^{-3}$ (red) and $\hat{u}_{r} = 10^{-4}$ (green). (b,c) Simulation results for the accumulation of (red) $a_{+}$ and (blue) $a_{-}$ for $\hat{u}_{r} = 10^{-2}$ and $\hat{u}_{r} = 10^{-3}$, respectively. The white curve represents the interface $a_{+} = a_{-}$.

channelization). As presented in figure 5a, the steady-state scalar field $\hat{h}$ distributions agree very well for three initial conditions. The obtained coupled networks of supply and drainage networks for $\hat{u}_{r} = 10^{-2}$ and $\hat{u}_{r} = 10^{-3}$ are also shown in figure 5b,c. For both cases, as expected, we obtain an equivalent distribution of both material densities with primary channels starting from boundaries having small defects in the spacing.

## 4.1.2. Effect of erosion and supply/consumption parameters

In this numerical experiment, we fix the exponents to unitary values ($m_{\pm} = n_{\pm} = 1$) and focus on how the role of parameters related to the relative rate of consumption ($r_{+}$) of the supplied material versus the generation rate ($r_{-}$) of the drained material that affects the feedback on the scalar field, which in turn affects the structure of the coupled networks. Keeping the value of all other parameters fixed, the change in these rates can be expressed as the modification in two non-dimensional channelization indices, $\mathcal{C}_{I_{+}}$ and $\mathcal{C}_{I_{-}}$.

The extent and spatial patterns of $a_{-}$ and $a_{+}$ for 55 cases with $\mathcal{C}_{I_{\pm}} \in [50, 100, \ldots, 500]$ and $\mathcal{C}_{I_{+}} \leq \mathcal{C}_{I_{-}}$ are analysed in this numerical study. Figure 6 shows the simulation results, where the supply network is represented in red (highlighting high-density region of $a_{+}$, i.e. $a_{+} > a_{-}$) and the drainage network is shown

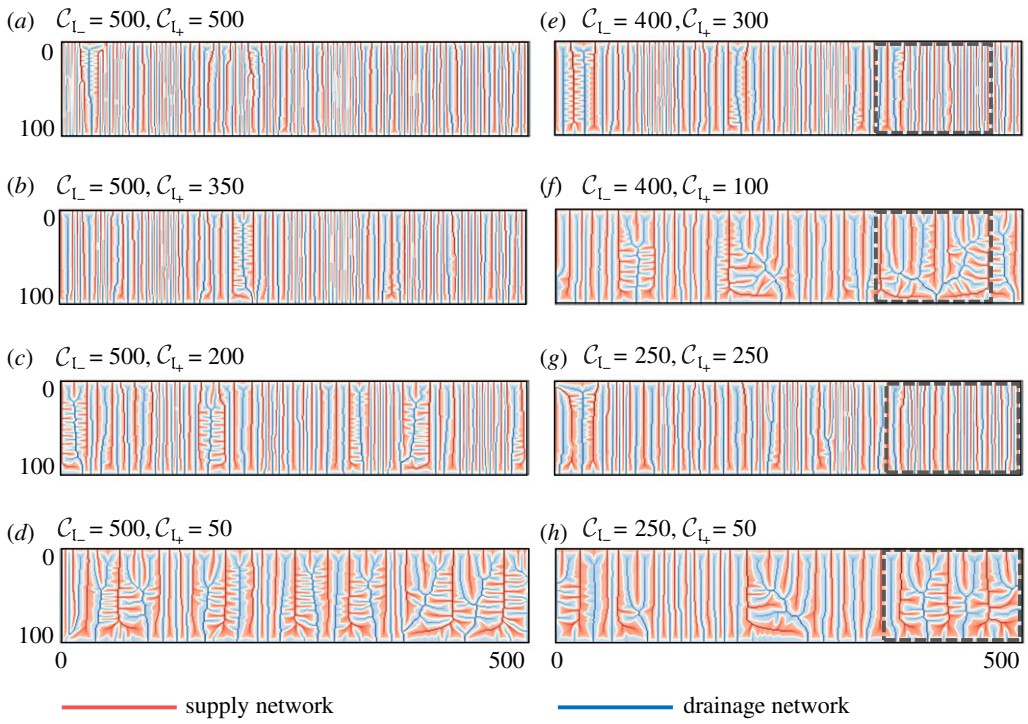

**Figure 6.** Simulation results for various values of $C_{I_+}$ and $C_{I_-}$ in a rectangular domain (width $= 100$, length $= 500$) with grid spacing $\Delta x = \Delta y = 1$. The accumulation of the input material $a_+$ is represented in red (highlighting $a_+ > a_-$), while the accumulation of the output material $a_-$ is shown in blue (highlighting $a_+ < a_-$). The white curve represents the interface $a_+ = a_-$ and separates the region dominated by the input material from the output material. The grey boxes in $(e,f,g,h)$ indicate the regions of the domain for which three-dimensional surface profiles of $h$ are shown in figure 7.

in blue (accentuating high-density region of $a_-$, i.e. $a_- > a_+$) with the white curve representing the interface $a_+ = a_-$. These spatial networks that evolve for various values of $C_{I_\pm}$ are quite distinctive, indicating the role of absolute as well as relative values of $C_{I_+}$ and $C_{I_-}$ on the overall pattern formation. Figure 6$a$,$b$ displays the plots where the number of channels of the supply and drainage network is high, with mostly straight channels and very little branching. Panels $(c,e,g)$ present the plots where comparatively less number of channels are observed with more branching. Panels $(d,f,h)$ show the plots where maximum branching is observed with curved channels compared with previous other cases.

The effect of the relative strength of $C_{I_+}$ and $C_{I_-}$ on the shape of the surface $h$ and hence on the spatial patterns of both networks is apparent from figure 7, which shows the three-dimensional surface plots of $h$ from the selected regions in figure 6. Panels $(a)$ and $(c)$ display the surface plots where the comparable opposing strength of $a_+$ and $a_-$ results in the formation of shallow ridge and valley patterns. Panels $(b)$ and $(d)$ show the surface plot for the branched region of $C_{I_+} = 100$, $C_{I_-} = 400$ and $C_{I_+} = 50$, $C_{I_-} = 250$, where high value of $C_{I_-}$ compared with $C_{I_+}$ results in branched channels of the networks with wide valleys and thin ridges at steady state.

The interplay between model parameters and boundary conditions becomes apparent when considering figure 6$f$,$g$. These cases have the same total $C_{I_+} + C_{I_-}$ for the two-field model, and thus, both solutions satisfy the same differential equations (2.7) and (2.8). Nevertheless, as is apparent in figure 6, the resulting branched structures are vastly different. This shows that the non-trivial boundary conditions allowed by the three-field model (as opposed to the two-field model) influence the solution throughout the domain, both quantitatively and qualitatively. A full discussion of this is included in appendix A.

The variety of patterns can be explained by the structure of equation (2.9) for the three-field model. Increasing the value of the respective channelization index enhances the feedback of supplied ($a_+$) and drained material ($a_-$) to actively form ridges and valleys of the two-dimensional 'landscape'. Pushing beyond the critical value of $C_{I_\pm}$ for the first channelization (3.5 in this case), we observe primary channels for both materials starting from the corresponding boundary (figure 4). As the value of $C_{I_\pm}$ increases, primary channels for both materials tend to form secondary branching, and so on. For a high and similar value of $C_{I_\pm}$, both networks have strong and comparable feedback to carve their

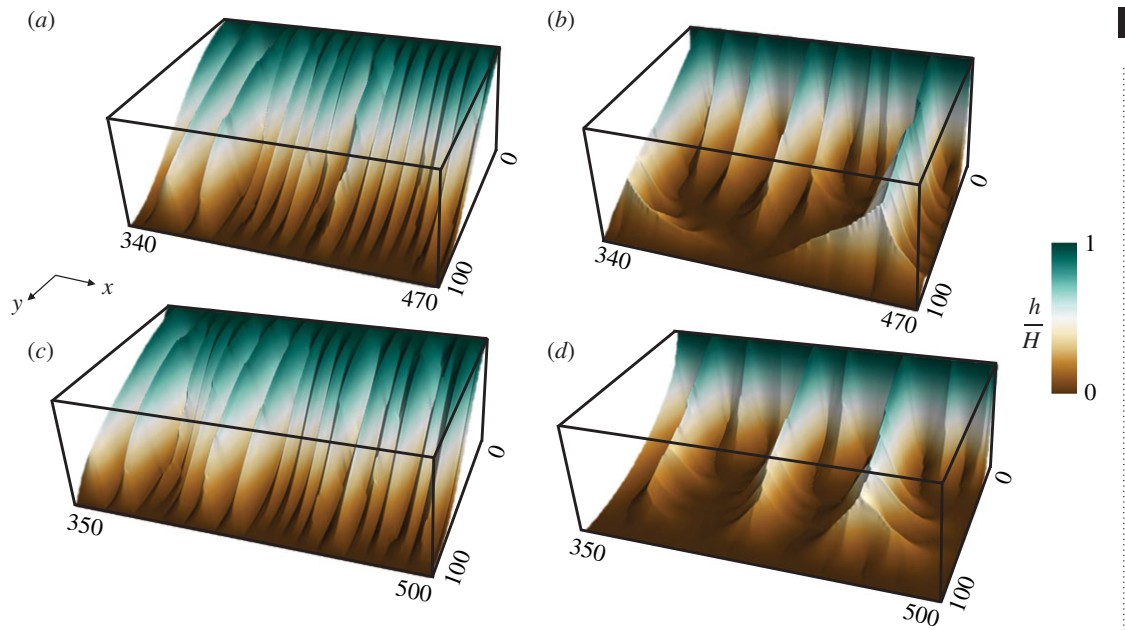

**Figure 7.** Three-dimensional surface profiles of $h$ for the regions highlighted in figure 6. (*a*) $\mathcal{C}_{I_-} = 400$, $\mathcal{C}_{I_+} = 300$. (*b*) $\mathcal{C}_{I_-} = 400$, $\mathcal{C}_{I_+} = 100$. (*c*) $\mathcal{C}_{I_-} = 250$, $\mathcal{C}_{I_+} = 250$. (*d*) $\mathcal{C}_{I_-} = 250$, $\mathcal{C}_{I_+} = 50$. Comparable values of $\mathcal{C}_{I_+}$ and $\mathcal{C}_{I_-}$ result in the formation of shallow ridges and valleys with less branching, while the disproportionate values of $\mathcal{C}_{I_+}$ and $\mathcal{C}_{I_-}$ result in wide and more branched valleys with sharp ridges.

respective networks by coalescing a large number of primary networks already formed to which the other material's network counteracts. This tendency of both networks to grow, competing against each other, makes them stuck in a large number of primary channels.

Reducing the value of $\mathcal{C}_{I_+}$ compared with $\mathcal{C}_{I_-}$ results in more branching as the primary channels of $a_-$ dominate and coalesce together to form branched patterns due to relatively smaller feedback from $a_+$. This effect of varying feedback of materials on the spatial patterns of coupled networks is shown in figure 8 by plotting the interface length $a_+ = a_-$ for $\mathcal{C}_{I_-} = 500$ and varying $\mathcal{C}_{I_+}$ from 500 to 50. For fixed value of $\mathcal{C}_{I_-} = 500$, decreasing the value $\mathcal{C}_{I_+}$ from 500 to 50 changes the spatial pattern with more branching and reduced number of stuck primary channels, as apparent from the reduced length of the interface $a_+ = a_-$.

Figure 9a reports the length of the interface $a_+ = a_-$ (denoted by $L_i$) for various values of $\mathcal{C}_{I_\pm}$. High values of $L_i$ occur for large and comparable values of $\mathcal{C}_{I_\pm}$, shown as the red region. Conversely, for disproportionate values of $\mathcal{C}_{I_\pm}$, the interface length ($L_i$) is smaller as shown in the blue region. We define a quantity $N_c$, which refers to the maximum number of either main supply or drainage channels of length greater than half of the width of the domain (50 in this case) originating from the boundaries of the domain. $N_c$ is plotted for 55 cases of various values of $\mathcal{C}_{I_\pm}$ as figure 9b, which looks similar to the plot of $L_i$ as expected. High values of $N_c$ occur for large and similar values of $\mathcal{C}_{I_\pm}$ again shown as the red region. More branching results in smaller number of main channels for $\mathcal{C}_{I_+} \ll \mathcal{C}_{I_-}$, as indicated by the blue in figure 9b. The scatter plot of $L_i$ versus $N_c$ with best-fit line having correlation coefficient $r = 0.988$ corroborates the close relationship between number of main channels and the interface length (figure 9c).

The simulation results shown in figure 6 can be mapped to different regions in the colour-plot of the contour length $L_c$. Figure 6a,b belong to the red region in figure 9a, where a high density of nearly unswerving main channels with a few offshoots is observed. For this reason, we refer to it as the congested region. Figure 6d,f,h exhibit the plots for the blue (branched) area in figure 9a, with heavily branched channels. Figure 6c,e,g display the plots from the yellow/green (transient) region, which lie within these two extremes.

### 4.1.3. Role of the exponents

Non-dimensionalization of the governing equations in §2.4 shows that the steady-state solution, for a fixed set of exponents in source and sink terms, can be described based on the absolute and relative values of two channelization indices. This has been verified in the previous section, where we obtain

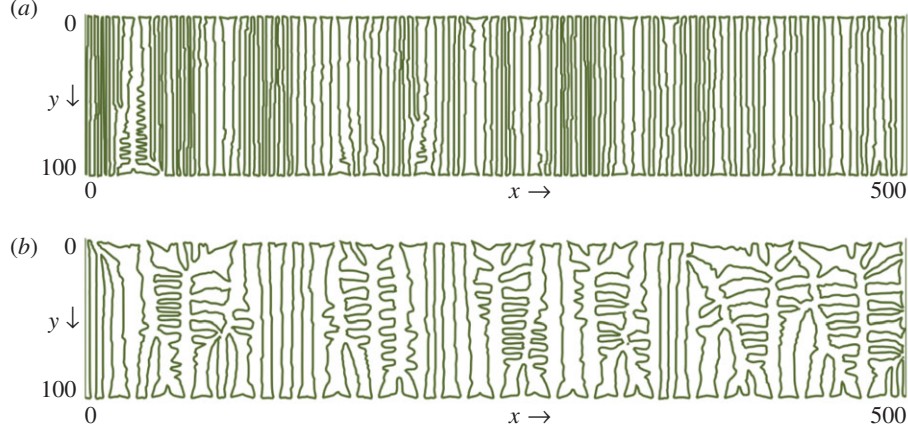

**Figure 8.** Plot of the interface $a_+ = a_-$ for $(\mathcal{C}_{l_+} = 500, \mathcal{C}_{l_-} = 500)$ and $(\mathcal{C}_{l_+} = 50, \mathcal{C}_{l_-} = 500)$.

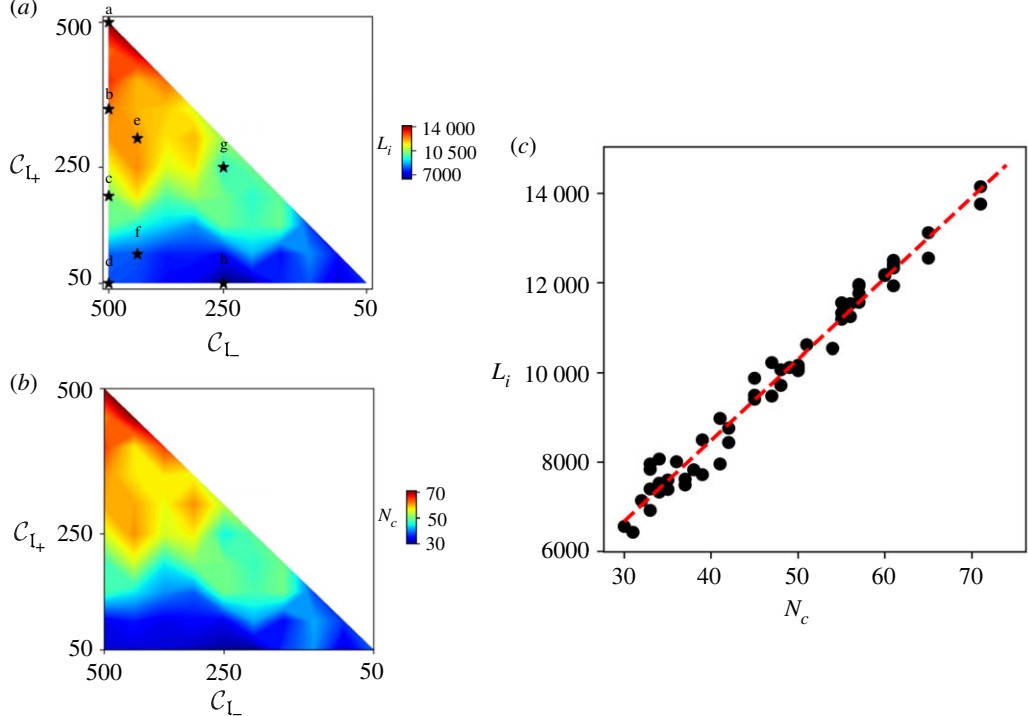

**Figure 9.** (a) Colour-plot of the interface length, $L_i$, for the values of $\mathcal{C}_{l_\pm} \in [50, 100, \ldots, 500]$ and $\mathcal{C}_{l_+} \leq \mathcal{C}_{l_-}$. Points a–h indicate the cases shown in figure 6. (b) Colour-plot of the number of main channels, $N_c$, for various values of $\mathcal{C}_{l_+}$ and $\mathcal{C}_{l_-} \in [50, 100, 150, \ldots 500]$. (c) Scatter plot of $L_c$ versus $N_c$ with the best-fit line (correlation coefficient $r = 0.988$).

a range of coupled networks as steady-state solutions that are classified by the value of both channelization indices for the unit value of exponents. In this section, we analyse the role of exponent values $m_\pm$ and $n_\pm$ on the model solutions.

In particular, we analyse the effect of $m_\pm$ and $n_\pm$ on the critical $\mathcal{C}_{I_\pm}$ value for channelization instability for a rectangular domain with a high aspect ratio (width $=100$, length $= 500$). Keeping the value of $m_\pm = 1$ and varying $n_\pm$ from 0.025 to 2.5, the value of critical $\mathcal{C}_{I_\pm}$ remains nearly constant in that range (blue points in figure 10).

We fix $n_\pm = 1$ in the next numerical experiment and observe the critical $\mathcal{C}_{I_\pm}$ for varying $m_\pm$ from 0.025 to 2.5. The critical $\mathcal{C}_{I_\pm}$ remains constant for $m_\pm$ between 1 and 2.5 and increases for lowering the value of $m_\pm$ below 1 (orange points in figure 10). For the range of $m_\pm$ from 0.025 to 0.5, there is a power-law scaling of critical $\mathcal{C}_{I_\pm}$ as shown in figure 10. This result indicates that the exponent of the gradient ($n_\pm$) in the

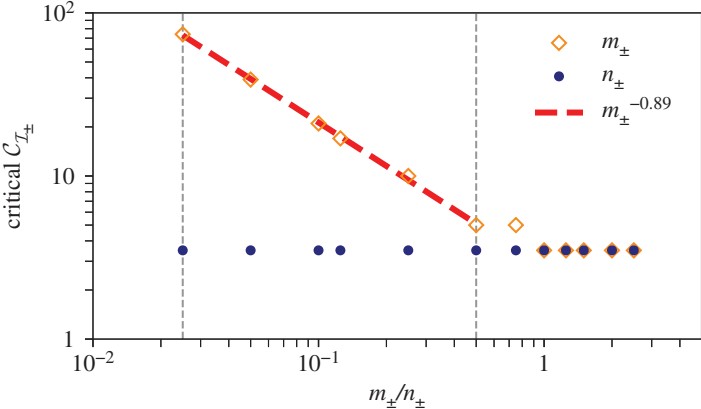

**Figure 10.** Dependence of critical $\mathcal{C}_{I_\pm}$ in a rectangular domain with high aspect ratio (width = 100, length = 500) with grid spacing $\Delta x = \Delta y = 1$. Blue points show the change of critical $\mathcal{C}_{I_\pm}$ on varying $n_\pm$ from 0.025 to 2.5 and keeping the value of $m_\pm = 1$. Orange points present the critical $\mathcal{C}_{I_\pm}$ on increasing $m_\pm$ from 0.025 to 2.5 for $n_\pm = 1$. The logarithmic plot of variation of the critical $\mathcal{C}_{I_\pm}$ on changing $m_\pm$ for fixed $n_\pm = 1$ reveals a power-law scaling between $m_\pm = 0.025$ and $m_\pm = 0.5$ with the power-law exponent $-0.89$ (obtained by fitting a (red) line in the range). Two vertical grey lines indicate the range of $m_\pm$ for the power-law scaling.

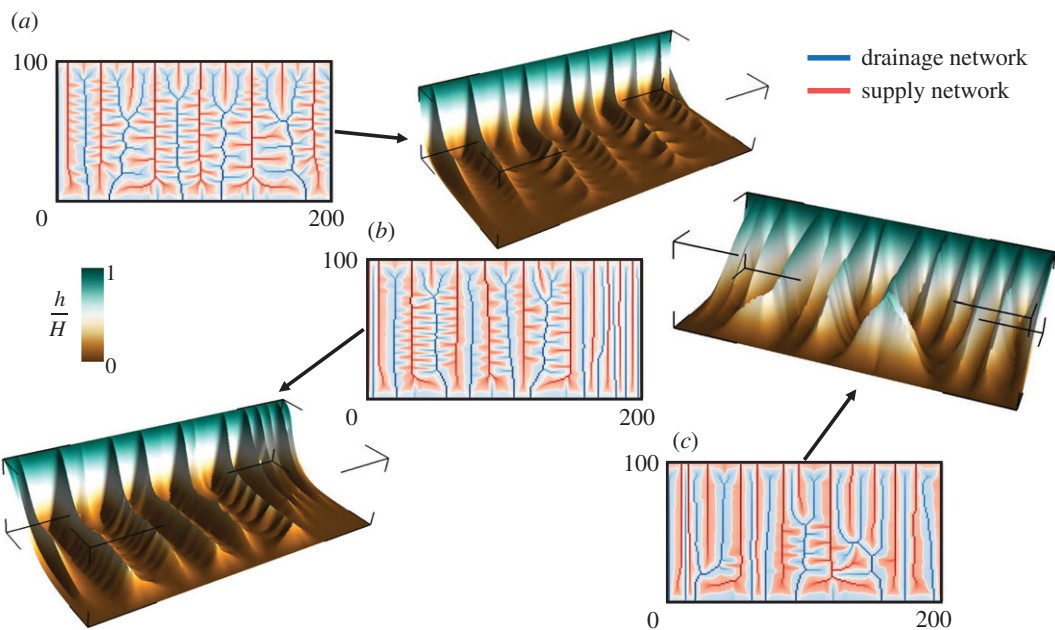

**Figure 11.** Steady-state solutions for $\mathcal{C}_{I_+} = 100$ and $\mathcal{C}_{I_-} = 400$ in a rectangular domain (width = 100, length = 200) with grid spacing $\Delta x = \Delta y = 1$ for $n_\pm = 1$ and (a) $m_\pm = 0.75$, (b) $m_\pm = 1.0$ (c) $m_\pm = 1.25$. The accumulation of $a_+$ is shown in red (highlighting $a_+ > a_-$), while the accumulation of the output material $a_-$ is presented in blue (highlighting $a_+ < a_-$). The white curve represents the interface $a_+ = a_-$. The corresponding three-dimensional surface profiles of the scalar field $h$ are shown as well.

sink/source term has little bearing on the first instance of channelization, while decreasing the exponent of the material densities ($m_\pm$) below one reduces the feedback of accumulated material density for channel formation in the domain, which is indicated by the high values of the critical $\mathcal{C}_{I_\pm}$.

Similarly to the spatial patterns between branched versus congested regime for the unit exponents of the source and sink terms, the solutions for non-unitary values of the exponents reflect an analogous spectrum of branched versus congested regime after the first channelization for a different range of $\mathcal{C}_{I_\pm}$. Figure 11 presents simulation results for the rectangular domain (width = 100, length = 200) varying the value of $m_\pm$ around one keeping $n_\pm = 1$, $\mathcal{C}_{I_+} = 100$ and $\mathcal{C}_{I_-} = 400$. The coupled supply and drainage networks are shown for $m_\pm$ 0.75 (<1), $m_\pm = 1.0$ and $m_\pm$ 1.25 (>1), where relatively more

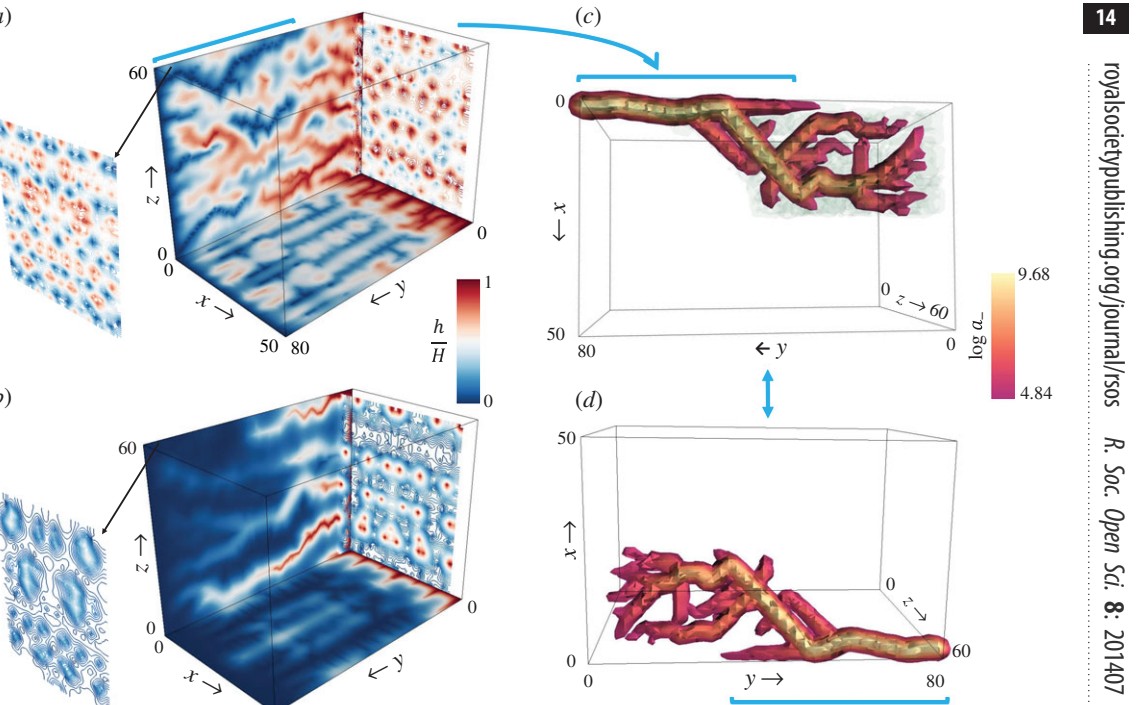

**Figure 12.** Simulation results for the three-dimensional domain ($x = 50$, $y = 80$, $z = 60$) with grid spacing $\Delta x = \Delta y = \Delta z = 1$. Contour plot presenting the scalar $h$ at the side boundaries and at the two cross-sections along the $y$-axis near the respective ends (top and bottom face) for ($a$) $\mathcal{C}_{I_+} = 1000$ and $\mathcal{C}_{I_-} = 1000$; ($b$) $\mathcal{C}_{I_+} = 200$ and $\mathcal{C}_{I_-} = 1000$. ($c$,$d$) The largest drainage conduit extracted in the domain for the case of $\mathcal{C}_{I_\pm} = 1000$ with light green coloured haze (in $c$) indicating the points in the domain from which the flow of drainage material is received in the conduit.

branched channels are observed in $m_\pm = 0.75$ compared with the networks in $m_\pm = 1.25$. The three-dimensional surface plots for these cases show that the thinnest ridges and deepest valleys are present in $m_\pm = 0.75$ (the scenario with the highest branching) among three solutions, which re-emphasizes the relationship between the shape of the field $h$ and the spatial patterns of coupled networks obtained in the domain.

## 4.2. Three-dimensional case

We apply the proposed model to a three-dimensional domain for a parallelepiped ($x = 50$, $y = 80$, $z = 60$), where $h$ now refers to a density field (can be viewed as a chemical signal's strength). There are fixed boundary conditions for two faces ($h(x, 0, z) = H = 10$ and $h(x, 80, z) = 0$) and zero Neumann boundary conditions at the remaining faces. $a_-$ is zero at $h(x, 0, z) = H = 10$ and $a_+$ is zero at $h(x, 80, z) = 0$, with closed boundary conditions in $a_\pm$ for the remaining four faces. We explore two cases keeping $\mathcal{C}_{I_-} = 1000$, while changing $\mathcal{C}_{I_+}$ from 1000 to 200 for $m_\pm = n_\pm = 1$. The simulation results are shown in figure 12, where the contour plots for the field $h$ are drawn on the two side faces (closed boundary conditions in $a_\pm$) along with contour line plots for the two cross-sections near the faces along the $y$-axis.

The steady-state solutions for the three-dimensional case agree with the patterns observed in the two-dimensional results. As shown in figure 12$a$, a large number of red contour curves (high-density region of $h$) in cross-section near the face $h(x, 0, z)$ extend over the domain and reach the cross-section near the face $h(x, 80, z)$, which is dominated by the blue contour curves (low-density region of $h$). This spatial pattern for $\mathcal{C}_{I_\pm} = 1000$ is similar to the solutions obtained in the two-dimensional case for the comparable value of two channelization indices, where a large number of shallow ridges and valleys, starting from either edge of the rectangular domain, propagate to the opposite end. We display the largest drainage conduit from the steady-state solution in figure 12$c$,$d$, where green haze in panel ($c$) indicates the points in the domain from which the flow is collected in the given conduit.

Similarly, the contour patterns for $\mathcal{C}_{I_+} = 200$ and $\mathcal{C}_{I_-} = 1000$ on the faces resemble the thin ridges and wide valleys obtained in the two-dimensional branched case when the values of $\mathcal{C}_{I_+}$ and $\mathcal{C}_{I_-}$ are

disproportionate (figure 12*b*). This is apparent as the tiny red contour curves (high-density region of *h*) in the cross-section near the face $h(x, 0, z)$ vanish in the cross-section near the face $h(x, 80, z)$ dominated by the blue contour curves (low-density region of *h*). This parallels the thin ridges that start from the fixed elevation end ($h = H$) and disappear near the fixed elevation end ($h = 0$) surrounded by deep and wide valleys, leading to the branched supply and drainage networks.

## 5. Conclusion

The minimalist model developed in this work leads to the formation of spatial patterns of combined supply and drainage networks in a continuous domain, whereby the corrugations of a mediating scalar field, *h*, cleave these competing networks in the same continuous domain. A channelization index ($\mathcal{C}_{I_\pm}$) corresponding to each material governs the relative intensity of the branching of these networks and the instability in the profile of *h*. The crucial role of the boundary condition for these coupled PDEs is particularly evident when reducing the presented three-field model to a two-field model for unit values of the exponents in source and sink terms, as the achieved simplification in the number of equations entails a complication in the boundary conditions, which is necessary to solve the same coexisting supply and drainage networks of the three-field model.

The role of different values of the two channelization indices on the shape of the scalar field *h* and the formation of coupled networks for $m_\pm = n_\pm = 1$ in equation (2.9) was explored along with the effect of non-unitary exponents on the first instant of channelization in a semi-infinite domain. As the specific patterns depend on these nonlinearities and the source and sink terms, future work will be devoted to adjusting them to cater to specific applications, as has been done in various other models, such as the minimalist versions of the well-known Keller–Segel model for chemotaxis [4,48] as well as the mechanochemical models of angiogenesis and vasculogenesis [2,42]. Concerning landscape evolution models, where *h* represents the elevation field of a natural landscape, a simple mathematical interpretation of the presented model can be a unique scenario where the tectonic uplift (source term) is a slope-dependent term that equals the erosion flux (sink term).

In the formulation of this minimalist model, we assume that the material flows along the direction of the steepest descent of field *h* with unit speed (equation (2.2)). There are network evolution models in geomorphology, hydrology, chemotaxis and vasculogenesis, where the flow has different spatio-temporal scales [2,4,59]. Different laws of material transport and diffusion properties can be employed in the presented numerical scheme for any loop-less flow-distribution network. The high adaptability of the numerical algorithm in [40] to any (structured/unstructured) grid can be attributed to its dependency on the node connectivity in a flow network rather than their spatial location in the discretized domain.

Initial and boundary conditions, as well as the domain geometry, are crucial in the proposed model. In particular, dislocation defects in the spatial patterns are expected to arise whenever the solution geometry does not match the domain geometry, as typical of pattern-forming systems [57,58]. These are evident in figure 4*c*,*d*, where the first-order networks form with small defects in the channel spacing. Increasing values of $\mathcal{C}_{I_\pm}$ drive the formation of complex networks, which are resolved up to the grid spacing. The reduced accuracy in the network approximation for heavy channelization is reflected on the imprint of the initial condition on the final steady-state solution for a very high value of channelization index, as the obtained channels get stuck due to the finite resolution of the discretized domain. Future work will be devoted to analysing the combined effect of these factors (geometry shape, grid resolution and alignment, initial and boundary conditions) on the uniqueness of numerical steady-state solutions in the proposed model.

The employed algorithm decreased the time complexity of the implicit solver by making the matrix system upper/lower triangular. This represented a crucial improvement in two-dimensional cases, where the memory requirement of the algorithm is not an issue [40]. However, the simulations in the three-dimensional domain require a large amount of memory, compared with the two-dimensional cases due to the increased input size of nodes and the corresponding auxiliary memory used by the algorithm during the execution. Since this increases the overall computational cost of the simulations in the three-dimensional cases, reducing the memory requirements of the numerical solver remains a priority so that the coupled patterns for a three-dimensional domain can be analysed in more depth.

Data accessibility. Data and relevant code for this research work are stored in GitHub: https://github.com/ShashankAnand1996/Supply_Drainage and have been archived within the Zenodo repository: https://doi.org/10.5281/zenodo.4435229.

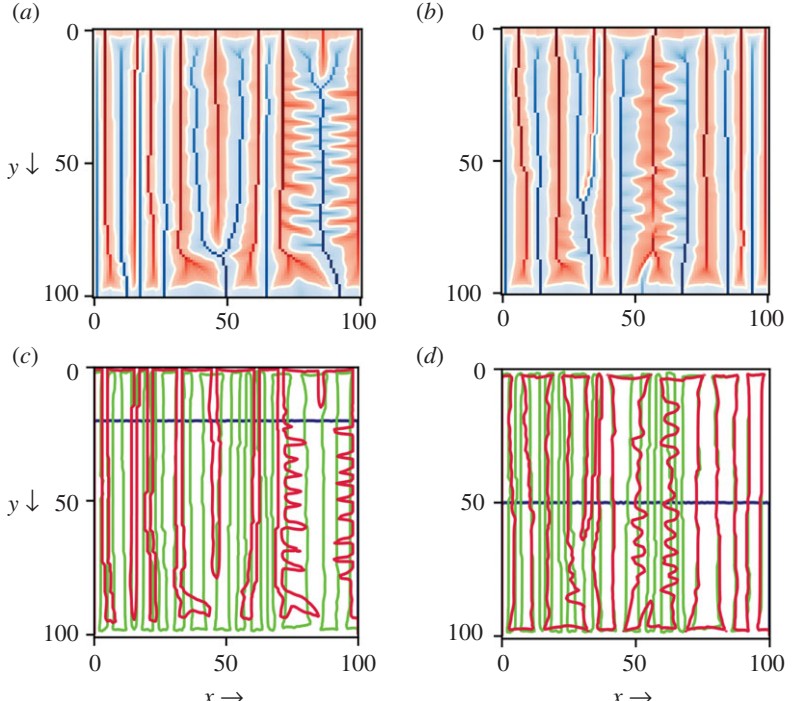

**Figure 13.** (a,b) Steady-state solutions for $\mathcal{C}_{I_+} = 100$, $\mathcal{C}_{I_-} = 400$ and $\mathcal{C}_{I_+} = 250$, $\mathcal{C}_{I_-} = 250$, respectively. The accumulation of $a_+$ is in red ($a_+ > a_-$), the accumulation of $a_-$ is in blue ($a_+ < a_-$) and the white curve is interface $a_+ = a_-$. (c,d) $a_* = 0$ at $t = 0$ (blue), $t =$ intermediate (green) and $t =$ steady state (red) for cases (a) and (b), respectively.

Authors' contributions. S.K.A. and A.P. designed the research; S.K.A. performed the research; S.K.A., M.H. and J.M.N. analysed the data. All authors discussed the results and wrote the paper.

Competing interests. We declare we have no competing interests.

Funding. No funding has been received for this article.

Acknowledgements. The authors acknowledge support from the US National Science Foundation (NSF) grant nos. EAR-1331846 and EAR-1338694, Innovation Award – Moore Science-to-Action Fund, and BP through the Carbon Mitigation Initiative (CMI) at Princeton University. A.P. and M.H. also acknowledge the support from the Princeton Institute for International and Regional Studies (PIIRS) and the Princeton Environmental Institute (PEI). J.M.N. was supported in part through Norwegian Research Council grant no. 250223. The authors are pleased to acknowledge that the simulations presented in this article were performed on computational resources managed and supported by Princeton Research Computing, a consortium of groups including the Princeton Institute for Computational Science and Engineering (PICSciE) and the Office of Information Technology's High Performance Computing Center and Visualization Laboratory at Princeton University. We thank Mario Putti and an anonymous reviewer for valuable comments and suggestions.

# Appendix A

In §2, we show that the original three-field model can be reduced to a two-field model consisting of equations (2.7) and (2.8) for the new spatial field, $a_*$, and the scalar field, $h$, under the assumption of unit exponents of the source and sink term in equation (2.3). These equations form a closed system where the dynamics of $h$ depends on the parameter $K_*$, which is determined by the summation of $r_+$ and $r_-$ only, instead of the two channelization indices that are defined for the three-field model.

We discuss here the dependency of complex boundary conditions of $a_*$ on the solution of spatial fields $a_+$ and $a_-$ by presenting steady-state solutions for a two-dimensional square domain with top edge ($y = 0$) at fixed high elevation ($H = 10$) and bottom edge ($y = L = 100$) at fixed zero elevation, with zero Neumann boundary conditions on the side edges. With the same values of $D = 10^{-3}$, $K = 10^{-5}$ and ($r_+ + r_-$) = 5, two cases are simulated as $r_+ = 1$, $r_- = 4$ ($\mathcal{C}_{I_+} = 100$, $\mathcal{C}_{I_-} = 400$) and $r_\pm = 2.5$ ($\mathcal{C}_{I_\pm} = 250$). Figure 13a,b shows the plot of steady-state supply and drainage material densities $a_+$ and $a_-$ for these cases. The difference in the obtained supply and drainage networks can be interpreted as the role of different values of $\mathcal{C}_{I_\pm}$ in the three-field model. However, the two cases correspond to the same value

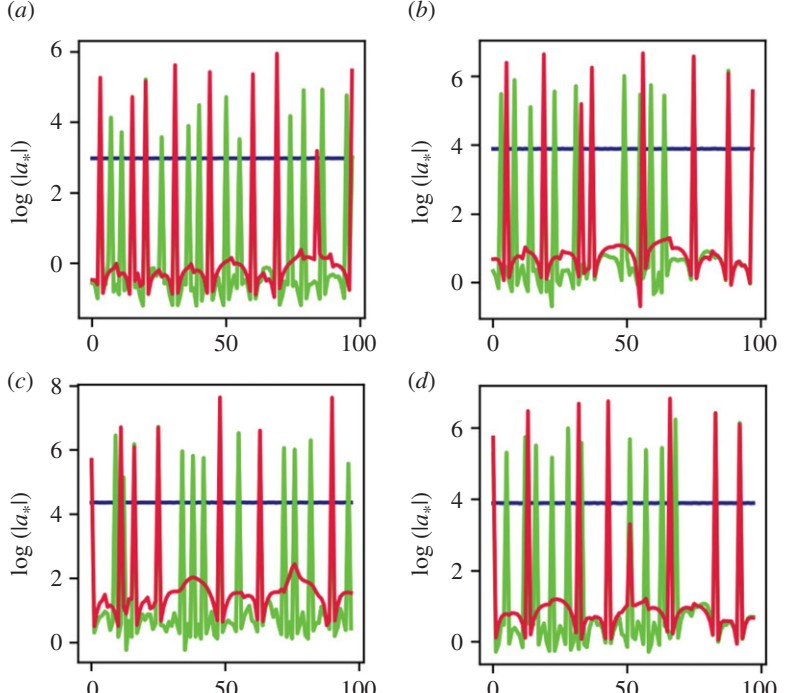

**Figure 14.** Blue ($t = 0$), green ($t = $ intermediate) and red ($t = $ steady state) curves represent the value of $a_*$ at the domain boundaries at different time steps. (a,b) $a_*$ at the top edge for $C_{l_+} = 100$, $C_{l_-} = 400$ and $C_{l_+} = 250$, $C_{l_-} = 250$, respectively. (c,d) $a_*$ at the bottom edge for $C_{l_+} = 100$, $C_{l_-} = 400$ and $C_{l_+} = 250$, $C_{l_-} = 250$, respectively.

of $K_* = 5 \times 10^{-5}$ for the two-field model, which indicates the crucial role of time-dependent boundary condition of $a_*$ on the obtained supply and drainage networks.

For the three-field model, the time-independent boundary conditions for $a_+$ is well defined, with $a_+ = 0$ at the bottom edge ($h = 0$) for both cases. Similarly, the boundary condition for $a_-$ is fixed in time throughout the simulations with $a_- = 0$ at the top edge of the square domain. For the two-field model, the boundary condition for $a_*$ in the two cases is different and is defined by specifying individual values of $r_+$ and $r_-$ initially, as shown in figure 13c,d. The blue curves for both cases, representing $a_* = 0$, vary in time, indicating the contribution of time-dependent boundary condition of $a_*$ on the simulation results.

This dependency is further shown in figure 14, where the value of $a_*$ at top and bottom edges of the square domain at different time steps are displayed for both cases. The different values of $a_*$ in time at domain boundaries indicate that the steady-state solutions with the same value of parameters ($K_* = 5 \times 10^{-5}$) are different because of the distinct time-varying boundary conditions for $a_*$. Therefore, the model can be simulated using the two fields of supply and drainage density with simple boundary conditions for the densities of the materials. In this way, the results can be interpreted in simple terms as the interplay of two indices of the supply and drainage density fields. If the two-field model is employed, the time-dependent boundary conditions for $a_*$ are extremely complex, and in practice the obtained coexisting networks can only be constructed from each of the two fields from which the sum $a_*$ originates.

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
