## [Peer Review File · Royal Society Open Science]

Review History

RSOS-201407.R0 (Original submission)

Review form: Reviewer 1 (Vaughan R. Voller)

Is the manuscript scientifically sound in its present form?

Yes

Are the interpretations and conclusions justified by the results?

Yes

Is the language acceptable?

Yes

Do you have any ethical concerns with this paper?

No

Have you any concerns about statistical analyses in this paper?

No

Recommendation?

Accept with minor revision (please list in comments)

Comments to the Author(s)

In this interesting and very worthwhile paper, the authors build on previous work related to the formation of channels in eroding drainage networks. Here, expanding on the previous equations and numerical tools, the authors look at how networks evolve in response to, in my understanding, erosion and deposition. Or more generally networks evolve in response to quantities (materials) that are removed (subtracted) from the system or supplied (added) to the system.

As I understand, the quantities added or subtracted from the system can be transported in two ways. In the first, a counter flowing case, the subtracted material moves down “slope” while the added material moves “up slope”. In the second transport case, referred to as co-flowing and the main focus of the work under review, both materials move down slope.

The resulting model is based on careful, consistent and verified developments. The authors explore an extensive parameter space, leading to a characterization of the generated channel patterns. In particular, they show how the number of channels, their “length”, and “branching” is controlled by two dimensionless parameters, measuring the relative strengths of sources and sinks. In addition, the authors also consider mathematical aspects of the boundary condition related to alternative model formulations – a 3 field vs. a 2 field model – and make an interesting initial foray in to the simulation of the formation of 3-D channels.

The current paper provides an excellent summary of the key theoretical concepts and consequences of the proposed channel forming model. The work may fall down a little, however, in connecting these models outcomes towards an understanding of the controls on physical channel networks. Thus, while I am very much in favor of this work, I would, as indicated by my comments below, like the authors to consider some points regarding the physical consequence of their model predictions.

1. Although this may have been explained in prior works, it may help readers here if the authors could provide brief explanations on

how the channel forming instability is manifest in the governing equations, and

how the flow routing employed in the numerical calculations, triggers this instability.

2. Might also be worthwhile if the authors summarized how the nature of the channels in co-flowing systems with erosion (removal) and deposition (supply) differ from those with just erosion ($CL+=0$). Does the observed suppression of the formation of branches, as $CL+$ approaches the value of $CL-$, have a physics interpretation?

3. The previous works from the authors look at systems driven by erosion countered by uplift. Is it reasonable to say that the current work essentially extends this by considering cases where erosion is countered by a nonlinear spacial/slope dependent uplift?

4. I realize that the model as proposed has a wide range of applications. But an interest of mine is in the “long -profile” formation in landscapes – from the uplift drainage collection in hill-slopes to the subsiding distribution in deltas. I have two questions in this regard.

In long profile models, the region of erosion (transport limited sediment production) is usually separated from the region of deposition (supply limited consumption). As I understand, in the current work, the processes of erosion and deposition compete at each point in the domain. Is this consistent with what happens in a landscape? I imagine, with the current model, that a more

abrupt transition from erosion to deposition could be induced by selection of the production/consumption rates in 2.1 and 2.4?

In the long-profile, it is understood that there are distinct differences between the morphology of drainage collection networks in the uplands and distribution networks into the ocean (considering one to be a simple mirror image of the other is often not appropriate). Does the current model-system have the potential of revealing why this might be the case.

5. Finally, on a numerical issue. As I understand the model is implemented on a structured row column finite-difference grid. Could the model be implemented on an unstructured grid (finite elements)? Might this provide more realistic networks that indicate confluence, branching, and bifurcation?

Smaller points

I think Figure 3 may be mislabeled. The red line is for $CL+=0$ and the green is for $CL-=0$.

Page 3, Line 45 -- In the above scenario

Review form: Reviewer 2 (Mario Putti)

Is the manuscript scientifically sound in its present form?

Yes

Are the interpretations and conclusions justified by the results?

Yes

Is the language acceptable?

Yes

Do you have any ethical concerns with this paper?

No

Have you any concerns about statistical analyses in this paper?

No

Recommendation?

Major revision is needed (please make suggestions in comments)

Comments to the Author(s)

Review of paper n. MRSOS-201407

"A minimalist model for co-evolving supply and drainage networks"

by

Shashank Anand, Milad Hooshyar, Jan Nordbotten, and Amilcare Porporato.

The paper describes a nonlinear model of supply/drainage network formation formed by three nonlinearly coupled PDEs for the mass density distribution of resources to be supplied/drained across the domain as transported by a sort of potential field as the flow of resources is proportional to the gradient of this scalar potential. The features of the derived model are explored by looking at the numerical solution of a few sample test cases using an ad-hoc numerical solver already developed by some of the same authors. e depending on the parameter

regimes (i.e. ratio between the resource input rates and diffusion) the time-asymptotic solution displays network-like structures that qualitatively resemble supply/drainage network pairs observed e.g. in river landscapes or a vascular systems.

The paper is well written and merits attention. I liked the paper and the ideas put forward and I think it is a very timely and beautiful work, in line with current trends in network formation models. I do have a few concerns/suggestions, which I list below and wish to be taken into consideration by the authors.

I will list first what I consider major points of concern, and then a few minor remarks and misprints.

Major points:

- p6, 16-15.

a. add the information that K and r are considered constant in space
 b. which putting the exponents m and n is later on only the case $m=n=1$ is explored? One has a lot of expectations to see what is happening for $m,n>1$ or $m,n<1$ but they are disappointed.
 c. More explanations on the rationale of why the evolution of the scalar field is governed by eq. (2.3) would be beneficial together with a discussion on the literature of these types of equations. Two papers with a mathematical flavor that report different reviews on these types come to mind:

1. Chen, A., Darbon, J., Buttazzo, G., Santambrogio, F., & Morel, J.-M. (2014). On the equations of landscape formation. *Interfaces and Free Boundaries*, 16(1), 105–136.

2. Haskovec, J., Markowich, P., Perthame, B., & Schlottbom, M. (2016). Notes on a PDE system for biological network formation. *Nonlinear Analysis*, 138, 127–155.

- p6, 121-27.

If my interpretation is correct, the authors should mention that this type of bcs allows only internal re-distribution of the densities, but these bcs are not used later on.

In any case, the overall discussion on boundary conditions is confusing. They are very important in determining the structure of the solution but their treatment is dispersed throughout the manuscript. I suggest to treat bcs comprehensively in one subsection. An example of this confusion, discussed also later on in my review, is that Figure 2 is the only place in the manuscript where the bcs of the first test case are mentioned, but the figure is never referenced in the text.

- p9, 144-47.

Here is one of my major concerns. The authors mention the addition of a (small) random noise to the initial conditions. It is never reference after this point, but I am mostly worried by it. Why is this done? What happens when this noise is zero? Do the branching structures form? Is this random noise responsible for the non-symmetries of the results? This must be explained. I guess that the dynamics of the proposed model is highly dependent upon initial conditions. If this is true, then the random component is fundamental to achieve the shown results, which are not as universal as the authors state. I am not saying that this observation mines the validity of the work. What I am saying is that the authors should discuss this and be transparent about it. There are a number of published models that have this behavior. It should be acknowledged and explored.

- p9, 147-56.

A longer description of the numerical approach would make the paper more self contained. I had to look at the paper [41] to understand how this is done. The sentence that the method is

"inspired from the notion of a flow network..." is hard to decipher. From reading in [41] I came up with the following intuition (is it correct?): a 5-point FD stencil is used for the discretization of the Laplacian and a sort-of upwind scheme is used for the gradient. The overall solution strategy seems a sort-of gradient descent algorithm where a time-splitting approach is used to accommodate diffusion. In any case, the solution strategy reminds me of the papers on the MAST approach proposed by Arico' and Tucciarelli some time ago (C Arico, T Tucciarelli, MAST solution of advection problems in irrotational flow fields, *Advances in water resources* 30 (3), 665-685).

- p11, fig5.

I do not understand the lack of symmetry. I would expect only straight lines for equal values of C_I . In the other cases, I would expect a recurring (periodic) pattern of branching structures. Why do these structures emerge only in the right portion of the domain? If this is the effect of the random component of the initial conditions, it should be thoroughly discussed, as it could be a symptom of instability of the numerical scheme or the model. In addition, the gradient descent algorithm could be trapped in some local minima away from the minimizer. Overall, it is a diffusion process and should maintain some sort of symmetry. Please discuss.

- p14, 145-52.

I am repeating and reinforcing here the point made in p11, fig5. I assume that also here the initial conditions are randomly perturbed. Again I am concerned by the fact that the lack of symmetry and the emergence of the preferential flow paths could be an undiscussed consequence of the random perturbation of the initial conditions. I'm always worried when these singular structures emerge as a consequence of randomness. It means that instabilities are present and not under control. Do you get the same results when no randomness is added to the initial conditions? If yes, then the final equilibrium is a stable equilibrium, if not, then the reader should be warned that the steady state equilibrium point is not unique (another equivalent way to put it is that the gradient descent algorithm is stopping in a local minimum which may be close to the global minimum but far from the global minimizer). In any case, in these situations numerical errors could be the triggers for the branching patterns. If this is the case, it should be discussed. In my view this is not necessarily detrimental to the manuscript. There are many models that behave like this as e.g. in your reference [12]. But in this situation the results should be analyzed using different strategies. I completely understand that this is beyond the scope of the current paper, but it should be mentioned to give a complete picture to the reader.

MINOR REMARKS:

- p4, 120-24. Here the authors introduce a scalar field but it is not clear what this scalar field is. Later on the authors assume that the transport velocity is proportional to the gradient of this scalar field and one may regard it as a potential field, and this could be anticipated here for better clarity;

- p4, 143. Awkward sentence. May be change "enter" in "entering"

- p5, eq. (2.2). The authors say "for simplicity" we assume... I am puzzled by this "simplicity" assumption. Simple as opposed to what? This is not simple at all and it is possibly this choice that sparks the richness of the solutions of the proposed model. I would like to see a few comments on this.

- p6, 17. Why the adjective "nonlocal"? Does it bear some significance in addition to "distributed"? If yes it should be explained, if not, then it should be avoided as it resembles other types of processes.

- p7, 142-50. Also here this discussion about bcs is confusing and does not add a lot of insights. Only conditions on top and bottom boundaries are discussed and set to homogeneous Dirichlet for the densities. If no flow is set on the other two boundaries then again there is only internal redistribution of densities, is it right?

- p7, eq. (2.7). This procedure is valid only if the production rates r are constant in space. Should be mentioned.

- p10, 18-16. Need more info on the simulation. For example, what is the grid size that allows the resolution of the ridges/valleys displayed in the figures? Does the solution change as the mesh is refined? I believe that grid-alignment problems should emerge. Has this been noticed?

- p10, 144-49. What are the BC's for a_+ and a_- ? I assume one and zero, but please specify. In a simpler way, you change the values of the C_I by changing only the consumption rates, right?

Awkward sentence at line 47: "Having all other parts parameters..."

- p11, 152. "as opposed two-field model" should be "as opposed to the two-field model"

- p11, 156. "hike" should be may be "hide"

- p11, can not  cannot

- p13, 17. "less number"  "smaller number"

- p14, 17-11. The sentence is convoluted. Please, try to rephrase to make it of more immediate understanding.

Decision letter (RSOS-201407.R0)

This year has been very difficult for everyone, and we want to take the opportunity to thank you for your continued support in 2020.

The Royal Society Open Science editorial office will be closed from the evening of Friday 18 December 2020 until Monday 4 January 2021. We will not be responding during this time. If you have received a deadline within this time period, please contact us as soon as possible to allow us to extend the deadline. If you receive any automated messages during this time asking you to meet a deadline, we offer apologies and invite you to respond after the festive period or during normal working hours.

With our best for a peaceful festive period and New Year, and we look forward to working with you in 2021.

Dear Mr Anand

The Editors assigned to your paper RSOS-201407 "A minimalist model for co-evolving supply and drainage networks" have now received comments from reviewers and would like you to revise the paper in accordance with the reviewer comments and any comments from the Editors. Please note this decision does not guarantee eventual acceptance.

Please submit your revised manuscript and required files (see below) no later than 21 days from today's (ie 15-Dec-2020) date. Note: the ScholarOne system will 'lock' if submission of the revision is attempted 21 or more days after the deadline. If you do not think you will be able to meet this deadline please contact the editorial office immediately.

on behalf of Dr Kenta Ishimoto (Associate Editor) and Mark Chaplain (Subject Editor)
openscience@royalsociety.org

Associate Editor Comments to Author (Dr Kenta Ishimoto):

Associate Editor: 1

Comments to the Author:

We appreciate the authors' patience for the time before the first decision. We look forward to your revised manuscript.

Reviewer comments to Author:

Reviewer: 1

Comments to the Author(s)

In this interesting and very worthwhile paper, the authors build on previous work related to the formation of channels in eroding drainage networks. Here, expanding on the previous equations and numerical tools, the authors look at how networks evolve in response to, in my understanding, erosion and deposition. Or more generally networks evolve in response to quantities (materials) that are removed (subtracted) from the system or supplied (added) to the system.

As I understand, the quantities added or subtracted from the system can be transported in two ways. In the first, a counter flowing case, the subtracted material moves down “slope” while the added material moves “up slope”. In the second transport case, referred to as co-flowing and the main focus of the work under review, both materials move down slope.

The resulting model is based on careful, consistent and verified developments. The authors explore an extensive parameter space, leading to a characterization of the generated channel patterns. In particular, they show how the number of channels, their “length”, and “branching” is controlled by two dimensionless parameters, measuring the relative strengths of sources and sinks. In addition, the authors also consider mathematical aspects of the boundary condition related to alternative model formulations – a 3 field vs. a 2 field model – and make an interesting initial foray in to the simulation of the formation of 3-D channels.

The current paper provides an excellent summary of the key theoretical concepts and consequences of the proposed channel forming model. The work may fall down a little, however, in connecting these models outcomes towards an understanding of the controls on physical channel networks. Thus, while I am very much in favor of this work, I would, as indicated by my comments below, like the authors to consider some points regarding the physical consequence of their model predictions.

1. Although this may have been explained in prior works, it may help readers here if the authors could provide brief explanations on

how the channel forming instability is manifest in the governing equations, and

how the flow routing employed in the numerical calculations, triggers this instability.

2. Might also be worthwhile if the authors summarized how the nature of the channels in co-flowing systems with erosion (removal) and deposition (supply) differ from those with just erosion ($CL+=0$). Does the observed suppression of the formation of branches, as $CL+$ approaches the value of $CL-$, have a physics interpretation?

3. The previous works from the authors look at systems driven by erosion countered by uplift. Is it reasonable to say that the current work essentially extends this by considering cases where erosion is countered by a nonlinear spacial/slope dependent uplift?

4. I realize that the model as proposed has a wide range of applications. But an interest of mine is in the “long -profile” formation in landscapes – from the uplift drainage collection in hill-slopes to the subsiding distribution in deltas. I have two questions in this regard.

In long profile models, the region of erosion (transport limited sediment production) is usually separated from the region of deposition (supply limited consumption). As I understand, in the current work, the processes of erosion and deposition compete at each point in the domain. Is this consistent with what happens in a landscape? I imagine, with the current model, that a more abrupt transition from erosion to deposition could be induced by selection of the production/consumption rates in 2.1 and 2.4?

In the long-profile, it is understood that there are distinct differences between the morphology of drainage collection networks in the uplands and distribution networks into the ocean (considering one to be a simple mirror image of the other is often not appropriate). Does the current model-system have the potential of revealing why this might be the case.

5. Finally, on a numerical issue. As I understand the model is implemented on a structured row column finite-difference grid. Could the model be implemented on an unstructured grid (finite elements)? Might this provide more realistic networks that indicate confluence, branching, and bifurcation?

Smaller points

I think Figure 3 may be mislabeled. The red line is for $CL+=0$ and the green is for $CL-=0$.

Page 3, Line 45 -- In the above scenario

Reviewer: 2

Comments to the Author(s)

Review of paper n. MRSOS-201407

"A minimalist model for co-evolving supply and drainage networks"

by

Shashank Anand, Milad Hooshyar, Jan Nordbotten, and Amilcare Porporato.

The paper describes a nonlinear model of supply/drainage network formation formed by three nonlinearly coupled PDEs for the mass density distribution of resources to be supplied/drained across the domain as transported by a sort of potential field as the flow of resources is proportional to the gradient of this scalar potential. The features of the derived model are explored by looking at the numerical solution of a few sample test cases using an ad-hoc numerical solver already developed by some of the same authors. Depending on the parameter regimes (i.e. ratio between the resource input rates and diffusion) the time-asymptotic solution displays network-like structures that qualitatively resemble supply/drainage network pairs observed e.g. in river landscapes or a vascular systems.

The paper is well written and merits attention. I liked the paper and the ideas put forward and I think it is a very timely and beautiful work, in line with current trends in network formation models. I do have a few concerns/suggestions, which I list below and wish to be taken into consideration by the authors.

I will list first what I consider major points of concern, and then a few minor remarks and misprints.

Major points:

- p6, 16-15.

a. add the information that K and r are considered constant in space

b. which putting the exponents m and n is later on only the case $m=n=1$ is explored? One has a lot of expectations to see what is happening for $m,n>1$ or $m,n<1$ but they are disappointed.

c. More explanations on the rationale of why the evolution of the scalar field is governed by eq. (2.3) would be beneficial together with a discussion on the literature of these types of equations. Two papers with a mathematical flavor that report different reviews on these types come to mind:

1. Chen, A., Darbon, J., Buttazzo, G., Santambrogio, F., & Morel, J.-M. (2014). On the equations of landscape formation. *Interfaces and Free Boundaries*, 16(1), 105-136.

2. Haskovec, J., Markowich, P., Perthame, B., & Schlottbom, M. (2016). Notes on a PDE system for biological network formation. *Nonlinear Analysis*, 138, 127-155.

- p6, 121-27.

If my interpretation is correct, the authors should mention that this type of bcs allows only internal re-distribution of the densities, but these bcs are not used later on.

In any case, the overall discussion on boundary conditions is confusing. They are very important in determining the structure of the solution but their treatment is dispersed throughout the manuscript. I suggest to treat bcs comprehensively in one subsection. An example of this confusion, discussed also later on in my review, is that Figure 2 is the only place in the manuscript where the bcs of the first test case are mentioned, but the figure is never referenced in the text.

- p9, 144-47.

Here is one of my major concerns. The authors mention the addition of a (small) random noise to the initial conditions. It is never reference after this point, but I am mostly worried by it. Why is this done? What happens when this noise is zero? Do the branching structures form? Is this random noise responsible for the non-symmetries of the results? This must be explained. I guess that the dynamics of the proposed model is highly dependent upon initial conditions. If this is true, then the random component is fundamental to achieve the shown results, which are not as universal as the authors state. I am not saying that this observation mines the validity of the work. What I am saying is that the authors should discuss this and be transparent about it. There are a number of published models that have this behavior. It should be acknowledged and explored.

- p9, 147-56.

A longer description of the numerical approach would make the paper more self contained. I had to look at the paper [41] to understand how this is done. The sentence that the method is "inspired from the notion of a flow network..." is hard to decipher. From reading in [41] I came up with the following intuition (is it correct?): a 5-point FD stencil is used for the discretization of the Laplacian and a sort-of upwind scheme is used for the gradient. The overall solution strategy seems a sort-of gradient descent algorithm where a time-splitting approach is used to accomodate diffusion. In any case, the solution strategy reminds me of the papers on the MAST approach proposed by Arico' and Tucciarelli some time ago (C Arico, T Tucciarelli, MAST solution of advection problems in irrotational flow fields, Advances in water resources 30 (3), 665-685).

- p11, fig5.

I do not understand the lack of symmetry. I would expect only straight lines for equal values of C_I . In the other cases, I would expect a recurring (periodic) pattern of branching structures. Why do these structures emerge only in the right portion of the domain? If this is the effect of the random component of the initial conditions, it should be thoroughly discussed, as it could be a symptom of instability of the numerical scheme or the model. In addition, the gradient descent algorithm could be trapped in some local minima away from the minimizer. Overall, it is a diffusion process and should maintain some sort of symmetry. Please discuss.

- p14, 145-52.

I am repeating and reinforcing here the point made in p11, fig5. I assume that also here the initial conditions are randomly perturbed. Again I am concerned by the fact that the lack of symmetry and the emergence of the preferential flow paths could be an undiscussed consequence of the random perturbation of the initial conditions. I'm always worried when these singular structures emerge as a consequence of randomness. It means that instabilities are present and not under control. Do you get the same results when no randomness is added to the initial conditions? If yes, then the final equilibrium is a stable equilibrium, if not, then the reader should be warned that the steady state equilibrium point is not unique (another equivalent way to put it is that the gradient descent algorithm is stopping in a local minimum which may be close to the global

minimum but far from the global minimizer). In any case, in these situations numerical errors could be the triggers for the branching patterns. If this is the case, it should be discussed. In my view this is not necessarily detrimental to the manuscript. There are many models that behave like this as e.g. in your reference [12]. But in this situation the results should be analyzed using different strategies. I completely understand that this is beyond the scope of the current paper, but it should be mentioned to give a complete picture to the reader.

MINOR REMARKS:

- p4, 120-24. Here the authors introduce a scalar field but it is not clear what this scalar field is. Later on the authors assume that the transport velocity is proportional to the gradient of this scalar field and one may regard it as a potential field, and this could be anticipated here for better clarity;

- p4, 143. Awkward sentence. May be change "enter" in "entering"

- p5, eq. (2.2). The authors say "for simplicity" we assume... I am puzzled by this "simplicity" assumption. Simple as opposed to what? This is not simple at all and it is possibly this choice that sparks the richness of the solutions of the proposed model. I would like to see a few comments on this.

- p6, 17. Why the adjective "nonlocal"? Does it bear some significance in addition to "distributed"? If yes it should be explained, if not, then it should be avoided as it resembles other types of processes.

- p7, 142-50. Also here this discussion about bcs is confusing and does not add a lot of insights. Only conditions on top and bottom boundaries are discussed and set to homogeneous Dirichlet for the densities. If no flow is set on the other two boundaries then again there is only internal redistribution of densities, is it right?

- p7, eq. (2.7). This procedure is valid only if the production rates r are constant in space. Should be mentioned.

- p10, 18-16. Need more info on the simulation. For example, what is the grid size that allows the resolution of the ridges/valleys displayed in the figures? Does the solution change as the mesh is refined? I believe that grid-alignment problems should emerge. Has this been noticed?

- p10, 144-49. What are the BC's for a_+ and a_- ? I assume one and zero, but please specify. In a simpler way, you change the values of the C_I by changing only the consumption rates, right?

Awkward sentence at line 47: "Having all other parts parameters..."

- p11, 152. "as opposed two-field model" should be "as opposed to the two-field model"

- p11, 156. "hike" should be may be "hide"

- p11, can not  cannot

- p13, 17. "less number"  "smaller number"

- p14, 17-11. The sentence is convoluted. Please, try to rephrase to make it of more immediate understanding.

===PREPARING YOUR MANUSCRIPT===

===PREPARING YOUR REVISION IN SCHOLARONE===

-- If you have uploaded ESM files, please ensure you follow the guidance at <https://royalsociety.org/journals/authors/author-guidelines/#supplementary-material> to include a suitable title and informative caption. An example of appropriate titling and captioning may be found at https://figshare.com/articles/Table_S2_from_Is_there_a_trade-off_between_peak_performance_and_performance_breadth_across_temperatures_for_aerobic_sc_ope_in_teleost_fishes_/3843624.

Author's Response to Decision Letter for (RSOS-201407.R0)

See Appendix A.

Decision letter (RSOS-201407.R1)

Dear Mr Anand,

It is a pleasure to accept your manuscript entitled "A minimalist model for co-evolving supply and drainage networks" in its current form for publication in Royal Society Open Science.

You can expect to receive a proof of your article in the near future. Please contact the editorial office (openscience@royalsociety.org) and the production office (openscience_proofs@royalsociety.org) to let us know if you are likely to be away from e-mail contact – if you are going to be away, please nominate a co-author (if available) to manage the proofing process, and ensure they are copied into your email to the journal.

on behalf of Dr Kenta Ishimoto (Associate Editor) and Mark Chaplain (Subject Editor)
openscience@royalsociety.org

Appendix A

A MINIMALIST MODEL FOR CO-EVOLVING SUPPLY AND DRAINAGE NETWORKS

RESPONSE TO THE REVIEWERS

Shashank Kumar Anand, Milad Hooshyar, Jan Martin Nordbotten and Amilcare Porporato

Manuscript ID: RSOS-201407

Manuscript Title: A minimalist model for co-evolving supply and drainage networks

Associate Editor Comments to Author (Dr Kenta Ishimoto)

Associate Editor: 1

Comments to the Author:

We appreciate the authors' patience for the time before the first decision. We look forward to your revised manuscript.

Thank you for giving us the opportunity to submit the revised manuscript. We have considered the suggestions of the reviewers to improve the presentation and better explain our results. We hope that the revised manuscript is now more suitable for publication.

Reviewer 1

In this interesting and very worthwhile paper, the authors build on previous work related to the formation of channels in eroding drainage networks. Here, expanding on the previous equations and numerical tools, the authors look at how networks evolve in response to, in my understanding, erosion and deposition. Or more generally networks evolve in response to quantities (materials) that are removed (subtracted) from the system

or supplied (added) to the system.

As I understand, the quantities added or subtracted from the system can be transported in two ways. In the first, a counter flowing case, the subtracted material moves down “slope” while the added material moves “up slope”. In the second transport case, referred to as co-flowing and the main focus of the work under review, both materials move down slope.

The resulting model is based on careful, consistent and verified developments. The authors explore an extensive parameter space, leading to a characterization of the generated channel patterns. In particular, they show how the number of channels, their “length”, and “branching” is controlled by two dimensionless parameters, measuring the relative strengths of sources and sinks. In addition, the authors also consider mathematical aspects of the boundary condition related to alternative model formulations—a 3 field vs. a 2 field model— and make an interesting initial foray in to the simulation of the formation of 3-D channels.

The current paper provides an excellent summary of the key theoretical concepts and consequences of the proposed channel forming model. The work may fall down a little, however, in connecting these models outcomes towards an understanding of the controls on physical channel networks. Thus, while I am very much in favor of this work, I would, as indicated by my comments below, like the authors to consider some points regarding the physical consequence of their model predictions.

We are glad that the reviewer found our work interesting. We have further improved the presentation by carefully revising it according to the reviewer’s suggestions.

1. Although this may have been explained in prior works, it may help readers here if the authors could provide brief explanations on how the channel forming instability is manifest in the governing equations, and how the flow routing employed in the numerical calculations, triggers this instability.

Thanks for this comment. We have modified the text to include it. In general, the feedback between drained/supplied material flow accumulation and the scalar field h is manifested via the sink/source term in Eq. (2.3). This can be explained clearly for the case of only erosion in a natural landscape.

The sink term indicates that erosion is high when we have high density of flow material and high magnitude of gradient of h . As we follow the direction of steepest descent and accumulate more drained material down the slope, and wherever accumulation is higher, we erode more. This feedback loop results in the trigger of channel instability, which is inhibited by diffusion process whose role is to smooth these instabilities. Hence, there is a threshold of channelization index above which these instabilities grow and result in the formation of valleys. For the case of detachment-limited erosion in semi-infinite domain, the linear stability analysis reveals that $C_{I_c} \approx 37$ is the critical channelization index, where first channel instability occurs [1]. In [2], we show that the presented numerical algorithm follows this theoretical prediction using D_∞ flow-direction method in the rectangular domain with high aspect ratio. We apply the same "modified breadth-first topological sort" algorithm here for the numerical calculation of both source and sink terms.

We have added one paragraph and modified an existing paragraph in the revised manuscript to explain the physical mechanism hidden in the form of sink/source term for channel forming instability in Section 2(b) of the revised manuscript (Lines 122-138):

A physical understanding of the feedback mechanism related to the source/sink term in Equation (2.3) can be achieved by inspecting the example of an eroding overland flow in a natural landscape. The sink term used in landscape evolution models (the same as that employed in Equation (2.3)) implies that heavy erosion of h occurs for large values of material density a_- and high magnitudes of h gradients [1, 3, 4]. Following the steepest descent direction of h , more accumulation of the drained material causes high erosion. This feedback loop in carving a preferential path creates a surface instability, which tends to be inhibited by the smoothing effect of diffusion. A threshold exists, above which the instability grows and results in the formation of complex valley network [1, 2].

In this model, the sink and source terms mathematically formalize the same conceptual framework shown in Figure 1, where the movement of materials carves out the preferential paths. Thus, for the 2-dimensional case, the scalar field h may be viewed as an elevation field of the ‘hypothetical’ landscape over which input and output materials move following Equation (2.5). As indicated by Equation (2.3), the accumulation of drainage material decreases the elevation that results in the formation of valleys (sink term). Conversely, the aggregation of the supply material increases the surface elevation that leads to the formation of ridges (source term). Consequently, the input material is accumulated on ridges, while the output material is concentrated in valleys.

2. Might also be worthwhile if the authors summarized how the nature of the channels in co-flowing systems with erosion (removal) and deposition (supply) differ from those with just erosion ($CL_+=0$). Does the observed suppression of the formation of branches, as CL_+ approaches the value of CL_- , have a physics interpretation?

We have carefully revised the manuscript to present the physical interpretation of the obtained spatial patterns in the model. For the landscape evolution models simulating first order processes in a natural landscape [5, 6, 1] with just detachment-limited erosion ($C_{T_+} = 0$) and positive uplift rate (source term), the ridges are created passively. This means that uplift influxes certain rate of sediment at each location in the domain and erosion create intricate network of valleys based on the feedback explained in the above point. Ridges are presented in the areas dominated by diffusion and low value of a_- and $|\nabla h|$, which are interlocked with actively formed valley network in the domain. On the other hand, in this model, the addition/deposition of material in the domain competes for the ridge formation actively with erosion/removal of material for valley formation.

Starting around a critical value of $C_{T_{\pm}}$ near the first channelization, we get primary channels starting from the boundary for both materials (Figure 4(b,c,d)). As the value of either index is increased, the tendency of primary channels is to blend together to create secondary branching, and so on. For a high and equivalent value of $C_{T_{\pm}}$, both networks have powerful and proportionate feedback to carve their respective networks by coalescing a large number of primary networks already formed to which the other material’s network counteracts. Therefore, this tendency of both networks

to grow, which are competing against each other, makes them stuck in a large number of primary channels and do not form branched spatial patterns.

We have changed two paragraphs in Section 4(a)(ii) to provide a better interpretation of the observed patterns (Lines 307-317 of the manuscript):

The variety of patterns can be explained by the structure of Equation (2.9) for the 3-field model. Increasing the value of the respective channelization index enhances the feedback of supplied (a_+) and drained material (a_-) to actively form ridges and valleys of the 2-dimensional ‘landscape’. Pushing beyond the critical value of $C_{\mathcal{I}\pm}$ for the first channelization (3.5 in this case), we observe primary channels for both materials starting from the corresponding boundary (Figure 4). As the value of $C_{\mathcal{I}\pm}$ increases, primary channels for both materials tend to form secondary branching, and so on. For a high and similar value of $C_{\mathcal{I}\pm}$, both networks have strong and comparable feedback to carve their respective networks by coalescing a large number of primary networks already formed to which the other material’s network counteracts. This tendency of both networks to grow, competing against each other, makes them stuck in a large number of primary channels.

(Lines 318-324):

Reducing the value of $C_{\mathcal{I}+}$ compared to $C_{\mathcal{I}-}$ results in more branching as the primary channels of a_- dominate and coalesce together to form branched patterns due to relatively smaller feedback from a_+ . This effect of varying feedback of materials on the spatial patterns of coupled networks is shown in Figure 8 by plotting the interface length $a_+ = a_-$ for $C_{\mathcal{I}-} = 500$ and varying $C_{\mathcal{I}+}$ from 500 to 50. For fixed value of $C_{\mathcal{I}-} = 500$, decreasing the value $C_{\mathcal{I}+}$ from 500 to 50 changes the spatial pattern with more branching and reduced number of stuck primary channels, as apparent from the reduced length of the interface $a_+ = a_-$.

3. The previous works from the authors look at systems driven by erosion countered by uplift. Is it reasonable to say that the current work essentially extends this by considering cases where erosion is countered by a nonlinear spacial/slope dependent uplift?

The reviewer is correct in pointing out that from a mathematical point of view, the source term in natural landscape evolution model has been replaced by a special slope dependent uplift for this mathematical model of coupled networks. We mention it in the updated manuscript (Lines 418-421):

Concerning landscape evolution models, where h represents the elevation field of a natural landscape, a simple mathematical interpretation of the presented model can be a unique scenario where the tectonic uplift (source term) is a slope dependent term that equals the erosion flux (sink term).

4. I realize that the model as proposed has a wide range of applications. But an interest of mine is in the “long-profile” formation in landscapes—from the uplift drainage collection in hill-slopes to the subsiding distribution

in deltas. I have two questions in this regard.

In long profile models, the region of erosion (transport limited sediment production) is usually separated from the region of deposition (supply limited consumption). As I understand, in the current work, the processes of erosion and deposition compete at each point in the domain. Is this consistent with what happens in a landscape? I imagine, with the current model, that a more abrupt transition form erosion to deposition could be induced by selection of the production/consumption rates in 2.1 and 2.4?

First of all, we would like to emphasize that our paper is not directly aimed at modeling landscapes, but more as a means to develop a minimalist model of simultaneous supply and drainage networks. However, the reviewer's comment regarding competition of erosion and deposition in landscapes is interesting. Following the literature on erosion and deposition flux over hill-slope, they both happen at every point, but the amount of either flux varies based on the landscape profile and the scale of the problem [7]. On a large spatial scale, as mentioned above by the reviewer, one can assume an idealization of high erosion flux in hill-slope regions compared to deposition in the low-lying delta. We agree with the reviewer that the model can function as having a very abrupt transition of consumption/deposition and production/erosion rates (Eq. 2.1). But the identical mathematical structure of source and sink term assumed in the model for minimalism and keeping the whole analysis general also needs to be modified according to the physics of the problem, such as deposition flux dependent on the mean concentration of particles and average vertical velocity, etc.

In principle, we can have spatially separated erosion and deposition; while this goes beyond our scope here, it will be interesting to analyze it in future contributions.

In the long-profile, it is understood that there are distinct differences between the morphology of drainage collection networks in the uplands and distribution networks into the ocean (considering one to be a simple mirror image of the other is often not appropriate). Does the current model-system have the potential of revealing why this might be the case.

As mentioned in our previous response, we agree with the reviewer that a mirror image system used in the model for source and sink terms can not be directly applied to the study of long profile formation in landscapes.

In this model, we assume a mirror-image behavior in the conceptualization of supply and drainage networks (Figure 1) to be the case rather than looking/justifying the distinct formation of both networks. This is evident from keeping the source and sink term in the same mathematical form in the governing equations as well as the flow equations of the two materials. We do this in the interest of developing a parsimonious model that can carve two networks in the same continuous domain using partial differential equations. The proposed model can be catered to the specific problem of interest such as chemotaxis, vasculogenesis, the long profile models (mentioned by the reviewer). Modifications can be done in the governing equation of materials' flow (adding diffusion, changing speed from unity, slope proportion, etc.), adjusting the mathematical structure of source and sink terms (including the exponents) based on the physics of the particular problem and spatial/temporal scales of description.

We mention this in Section (5) of the revised manuscript (Lines 414-418):

As the specific patterns depend on these nonlinearities and the source and sink terms, future work will be devoted to adjusting them to cater to specific applications, as has been done in various other models, such as the minimalist versions of the well-known Keller–Segel model for chemotaxis [8, 9] as well as the mechanochemical models of angiogenesis and vasculogenesis [10, 11].

and (Lines 422-427):

In the formulation of this minimalist model, we assume that the material flows along the direction of the steepest descent of field h with unit speed (Equation (2.2)). There are network evolution models in geomorphology, hydrology, chemotaxis and vasculogenesis, where the flow has different spatiotemporal scales [8, 10, 12]. Different laws of material transport and diffusion properties can be employed in the presented numerical scheme for any loop-less flow-distribution network.

5. Finally, on a numerical issue. As I understand the model is implemented on a structured row column finite-difference grid. Could the model be implemented on an unstructured grid (finite elements)? Might this provide more realistic networks that indicate confluence, branching, and bifurcation?

Thanks for this remark. We have included it in the revised manuscript. The numerical algorithm from [2] employed in the proposed model for the sink/source term is adaptable to any type of discrete mesh (Raster/Triangulated irregular network/Any other irregular grid) as it traverses based on the connectivity of the nodes in the flow network. It does not depend on the spatial location of the node in the domain. So, given a flow-network in the spatial domain, it can generate the linear layout which can be followed to update every point in the domain implicitly and efficiently. The overall accuracy of the solutions in the modified algorithm will depend on various factors such as grid resolution, numerical method accuracy, flow-direction method used to approximate the network in structured/unstructured grid for both materials, etc. which have to be carefully verified as has been done for this scheme in [2].

We mention it in the updated manuscript (Lines 425-429):

Different laws of material transport and diffusion properties can be employed in the presented numerical scheme for any loop-less flow-distribution network. The high adaptability of the numerical algorithm in [2] to any (structured/unstructured) grid can be attributed to its dependency on the node connectivity in a flow network rather than their spatial location in the discretized domain.

Smaller points:

I think Figure 3 may be mislabeled. The red line is for $CL_+=0$ and the green is for $CL_-=0$.

Thank you. The figure was mislabelled in the original manuscript. It has been corrected in the updated manuscript.

Figure R1: (a,b): Steady-state solutions given by Equations (3.2) and (3.3) for three cases of $C_{I_+} = 0$ (red), $C_{I_-} = 0$ (green), and $C_{I_{\pm}} = 25$ (blue).

Page 3, Line 45— In the above scenario

We have corrected the sentence in the revised manuscript:

"Reversing the flow direction of the density a_+ in the above scenario, the problem can be formulated ..."

Reviewer 2

The paper describes a nonlinear model of supply/drainage network formation formed by three nonlinearly coupled PDEs for the mass density distribution of resources to be supplied/drained across the domain as transported by a sort of potential field as the flow of resources is proportional to the gradient of this scalar potential. The features of the derived model are explored by looking at the numerical solution of a few sample test cases using an ad-hoc numerical solver already developed by some of the same authors. Depending on the parameter regimes (i.e. ratio between the resource input rates and diffusion) the time-asymptotic solution displays network-like structures that qualitatively resemble supply/drainage network pairs observed e.g. in river landscapes or a vascular systems.

The paper is well written and merits attention. I liked the paper and the ideas put forward and I think it is a very timely and beautiful work, in line with current trends in network formation models.

We thank the reviewer for assessing our work. We are glad that the paper was found to be clearly written and in line with the current work on network formation. We have strived to further improve its presentation by carefully revising it according to the suggestions.

I do have a few concerns/suggestions, which I list below and wish to be taken into consideration by the authors. I will list first what I consider major points of concern, and then a few minor remarks and misprints.

Major points:

p6, 16-15.

a. add the information that K and r are considered constant in space

We now mention this explicitly, as it provides better clarity for the results presented in the manuscript (Lines 102-104):

where D is the diffusion coefficient, $K > 0$, $m_{\pm} > 0$ and $n_{\pm} > 0$ are model parameters and r_{\pm} indicate production rates for the respective material. In this work, we keep these parameters constant over the whole spatial domain.

b. which putting the exponents m and n is later on only the case $m=n=1$ is explored? One has a lot of expectations to see what is happening for $m,n>1$ or $m,n<1$ but they are disappointed.

Thanks for this comment. We have added a new section on the effect of varying values of m_{\pm} and n_{\pm} on the obtained solutions. Based on numerical simulations, we discuss the ‘critical’ value of $\mathcal{C}_{\mathcal{I}_{\pm}}$ that marks the onset of channelization in the domain for non-unity values of the either exponent. We also present a set of simulation results in the section while varying m_{\pm} for unit value of n_{\pm} when the value of two channelization indices are disproportionate (Lines 342-371):

Role of the exponents

Non-dimensionalization of the governing equations in Section 2(d) shows that the steady-state solution, for a fixed set of exponents in source and sink terms, can be described based on the absolute and relative values of two channelization indices. This has been verified in the previous section, where we obtain a range of coupled networks as steady-state solutions that are classified by the value of both channelization indices for the unit value of exponents. In this section, we analyze the role of exponent values m_{\pm} and n_{\pm} on the model solutions.

Figure R2: Dependence of critical $\mathcal{C}_{\mathcal{I}_{\pm}}$ in a rectangular domain with high aspect ratio (width = 100, length = 500) with grid spacing $\Delta x = \Delta y = 1$. Blue points show the change of critical $\mathcal{C}_{\mathcal{I}_{\pm}}$ on varying n_{\pm} from 0.025 to 2.5 and keeping the value of $m_{\pm} = 1$. Orange points present the critical $\mathcal{C}_{\mathcal{I}_{\pm}}$ on increasing m_{\pm} from 0.025 to 2.5 for $n_{\pm} = 1$. The logarithmic plot of variation of the critical $\mathcal{C}_{\mathcal{I}_{\pm}}$ on changing m_{\pm} for fixed $n_{\pm} = 1$ reveals a power-law scaling between $m_{\pm} = 0.025$ and $m_{\pm} = 0.5$ with the power-law exponent -0.89 (obtained by fitting a (red) line in the range). Two vertical gray lines indicate the range of m_{\pm} for the power-law scaling.

In particular, we analyze the effect of m_{\pm} and n_{\pm} on the critical $\mathcal{C}_{\mathcal{I}_{\pm}}$ value for channelization instability for a rectangular domain with a high aspect ratio (width = 100, length = 500). Keeping the value of $m_{\pm} = 1$ and varying n_{\pm} from 0.025 to 2.5, the value of critical $\mathcal{C}_{\mathcal{I}_{\pm}}$ remains nearly constant in that range (blue points in Figure R2).

We fix $n_{\pm} = 1$ in the next numerical experiment and observe the critical $\mathcal{C}_{\mathcal{I}_{\pm}}$ for varying m_{\pm} from 0.025 to 2.5. The critical $\mathcal{C}_{\mathcal{I}_{\pm}}$ remains constant for m_{\pm} between 1 and 2.5 and increases for lowering the value of m_{\pm} below 1 (orange points in Figure R2). For the range of m_{\pm} from 0.025 to 0.5, there is a power-law scaling of critical $\mathcal{C}_{\mathcal{I}_{\pm}}$ as shown in Figure R2. This result indicates that the exponent of the gradient (n_{\pm}) in the sink/source term has little bearing on the first instance of channelization, while decreasing the exponent of the material densities (m_{\pm}) below one reduces the

feedback of accumulated material density for channel formation in the domain, which is indicated by the high values of the critical $\mathcal{C}_{\mathcal{I}_{\pm}}$.

Figure R3: Steady-state solutions for $\mathcal{C}_{\mathcal{I}_+} = 100$ and $\mathcal{C}_{\mathcal{I}_-} = 400$ in a rectangular domain (width = 100, length = 200) with grid spacing $\Delta x = \Delta y = 1$ for $n_{\pm} = 1$ and (a): $m_{\pm} = 0.75$ (b): $m_{\pm} = 1.0$ (c): $m_{\pm} = 1.25$. The accumulation of a_+ is shown in red (highlighting $a_+ > a_-$), while the accumulation of the output material a_- is presented in blue (highlighting $a_+ < a_-$). The white curve represents the interface $a_+ = a_-$. The corresponding 3-dimensional surface profiles of the scalar field h are shown as well.

Similarly to the spatial patterns between branched versus congested regime for the unit exponents of the source and sink terms, the solutions for non-unitary values of the exponents reflect an analogous spectrum of branched versus congested regime after the first channelization for a different range of $\mathcal{C}_{\mathcal{I}_{\pm}}$. Figure R3 presents simulation results for the rectangular domain (width = 100, length = 200) varying the value of m_{\pm} around one keeping $n_{\pm} = 1$, $\mathcal{C}_{\mathcal{I}_+} = 100$ and $\mathcal{C}_{\mathcal{I}_-} = 400$. The coupled supply and drainage networks are shown for $m_{\pm} = 0.75$ (< 1), $m_{\pm} = 1.0$, and $m_{\pm} = 1.25$ (> 1), where relatively more branched channels are observed in $m_{\pm} = 0.75$ compared to the networks in $m_{\pm} = 1.25$. The 3-dimensional surface plots for these cases show that thinnest ridges and deepest valleys are present in $m_{\pm} = 0.75$ (the scenario with the highest branching) among three solutions, which reemphasizes the relationship between the shape of the field h and the spatial patterns of coupled networks obtained in the domain.

c. More explanations on the rationale of why the evolution of the scalar field is governed by eq. (2.3) would be beneficial together with a discussion on the literature of these types of equations. Two papers with a mathematical flavor that report different reviews on these types come to mind:

1. Chen, A., Darbon, J., Buttazzo, G., Santambrogio, F., Morel, J.-M. (2014). On the equations of landscape formation. *Interfaces and Free Boundaries*, 16(1), 105–136.
2. Haskovec, J., Markowich, P., Perthame, B., Schlottbom, M. (2016). Notes on a PDE system for biological network formation. *Nonlinear Analysis*, 138, 127–155.

Thank you for this remark. We have included a discussion on the rationale of selecting Equation (2.3) in the model and mentioned the literature that reviews these types of non-linear pattern-forming equations (including the above-mentioned papers). We now explain the crucial feedback mechanism possible by the mathematical form of the sink/source term employed in the model for carving the preferential path as has been observed for drainage network in landscape evolution models. The same kind of equations employed in hydrogeomorphology, chemotaxis, vasculogenesis, atomic surface growth models, etc. are also discussed in the revised text (Lines 122-130):

A physical understanding of the feedback mechanism related to the source/sink term in Equation (2.3) can be achieved by inspecting the example of an eroding overland flow in a natural landscape. The sink term used in landscape evolution models (the same as that employed in Equation (2.3)) implies that heavy erosion of h occurs for large values of material density a_- and high magnitudes of h gradients [1, 3, 4]. Following the steepest descent direction of h , more accumulation of the drained material causes high erosion. This feedback loop in carving a preferential path creates a surface instability, which tends to be inhibited by the smoothing effect of diffusion. A threshold exists, above which the instability grows and results in the formation of complex valley network [1, 2].

(Lines 146-150):

The mathematical structure of the proposed model resembles complex models of drainage network evolution, vasculogenesis, chemotaxis, and in general, biological network-formation models as well as surface-growth models [13, 14, 11, 15, 16, 17, 18, 19]. Specifically, the core component of the model resembles minimalist versions of the well-known Keller–Segel model for chemotaxis under negligible diffusion of biological cells [9, 20, 21].

- p6, l21-27.

If my interpretation is correct, the authors should mention that this type of bcs allows only internal redistribution of the densities, but these bcs are not used later on. In any case, the overall discussion on boundary conditions is confusing. They are very important in determining the structure of the solution but their treatment is dispersed throughout the manuscript. I suggest to treat bcs comprehensively in one subsection. An example of this confusion, discussed also later on in my review, is that Figure 2 is the only place in the manuscript where the bcs of the first test case are mentioned, but the figure is never referenced in the text.

We agree with the reviewer that the discussion of boundary conditions should be comprehensive and not scattered. We have added a new section (Section 2(c)) in the revised manuscript. We now more clearly state the crucial role of boundary conditions for the internal-distribution of material densities, marking entry and exit points of 2-dimensional

and 3-dimensional domains discussed in the manuscript. We also reference Figure 2, which got missed in the original manuscript due to typo in the figure label (Lines 151-189):

Boundary conditions

The boundary conditions play a crucial role in obtaining solutions for the internal distribution of the densities over the domain. We consider 2-dimensional and 3-dimensional domains in the shape of a rectangle or parallelepiped, respectively, with the top edge/face (Ω_t) at a fixed higher value ($h = H$) compared to the bottom edge/face (Ω_b) at a fixed lower value ($h = 0$). The remaining side edges/faces (Ω_s) follow zero Neumann boundary conditions in h , which provide closed boundary conditions in the densities of the materials ($a_{\pm} = 0$). For Problem I, the proposed arrangement induces a directionality to the movement of the two materials in the domain with top (Ω_t) and bottom (Ω_b) edges/faces functioning as the exit boundaries for a_+ and a_- , respectively. Assuming that the densities of the two materials are negligible at their upstream domain boundaries, the boundary conditions become simple and time-independent as $a_+(\Omega_b) = a_-(\Omega_t) = 0$. Under such boundary conditions and the assumption of spatially uniform production rates (r_+ and r_-), the governing equations compute the counter flow of the materials across the domain (including at the boundaries where the densities are not specified i.e., $a_+(\Omega_t)$ and $a_-(\Omega_b)$). Figure R4 illustrates the boundary conditions used in this work as well as the corresponding solutions near the first channel instability in the 2-dimensional case

For Problem II, the boundary conditions for h are the same as for Problem I, with top (Ω_t) and bottom (Ω_b) edges/faces functioning now as the entry and exit boundaries for the domain, respectively. Under the assumption that no drainage material exits from Ω_t and no supply material is conveyed out of Ω_b , the boundary conditions of the material densities remain simple and time-independent as $a_+(\Omega_b) = a_-(\Omega_t) = 0$. The zero Neumann boundary conditions in h on side edges correspond to closed boundary conditions for both material densities ($a_{\pm} = 0$). Under these conditions, the governing equations determine the flow and distribution of supplied and drained materials over the spatial domain (including at the boundaries where the densities are not specified i.e., $a_+(\Omega_t)$ and $a_-(\Omega_b)$) under the assumption of spatially uniform consumption (r_+) and production (r_-) rates.

It is interesting to observe that the model can be reduced to a 2-field system for the case $m_{\pm} = n_{\pm} = 1$ and constant r_{\pm} over the spatial domain. Multiplying the continuity equations for a_+ and a_- (Equation (2.4)) by r_+ and r_- respectively, and subtracting, one can write the single equation for a new spatial field

$$a_* = \frac{r_+ a_+ - r_- a_-}{r_+ + r_-}, \quad (\text{R1})$$

as

$$-\nabla \cdot \left(a_* \frac{\nabla h}{|\nabla h|} \right) = -1. \quad (\text{R2})$$

Equation (2.3) then can be re-written using Equations (R1) and (R2) as

$$\frac{\partial h}{\partial t} = D \nabla^2 h + K_* a_* |\nabla h|, \quad (\text{R3})$$

Figure R4: Schematic representation of the boundary conditions used in the model. (a): Surface profile of the scalar field h in a portion of the rectangular domain near the first channel instability, where H is the maximum value in the domain (see part 4(a) for details). Three shallow ridges and two shallow valleys can be observed in the plotted profile. (c): Boundary conditions of a_+ and a_- counter-flow drainage problem (Problem I) and co-flow supply and drainage problem (Problem II). Two (red) channels of a_+ and three (blue) channels of a_- corresponding to the ridges and valleys in panel (a) are observed. The white curve, representing the interface $a_+ = a_-$, separates the regions dominated by each material. Four contour lines of the scalar field h are plotted along with black streamlines which indicate the flow direction of the materials. (b,d): Obtained signals of a_+ and a_- at Ω_t and Ω_b , respectively, with peaks indicating channel formation at the corresponding domain boundaries.

where $K_* = K(r_+ + r_-)$. Equations (R2) and (R3) form a 2-field equivalent formulation (a_*, h) to the proposed 3-field model (a_+, a_-, h) for unit values of the exponents in Equation (2.3). The achieved simplification is, however, only apparent, as the knowledge of a_+ and a_- is required in advance to obtain the boundary conditions of the new spatial field a_* in the reduced model corresponding to the solution of the 3-field model with the time-independent boundary conditions. We refer to Appendix 6 for a detailed discussion of the boundary conditions for the 2-field model in the 2-dimensional case.

- p9, 144-47.

Here is one of my major concerns. The authors mention the addition of a (small) random noise to the initial conditions. It is never reference after this point, but I am mostly worried by it. Why is this done? What happens when this noise is zero? Do the branching structures form? Is this random noise responsible for the non-symmetries of the results? This must be explained.

Thanks for this comment; we carefully clarified this point in the revised version. In a previous paper, it was shown that the analytical un-channelized solution for the proposed model (Eq. 3.1) in a semi-infinite domain as well as in the

original model of landscape evolution is unstable to small perturbation above a critical value of parameters. This is discussed in [1], where the numerical algorithm's results (the same used in this model) are compared with the analytical predictions of the original landscape evolution model (LEM). In LEM, the model has only erosion ($C_{\mathcal{I}_+} = 0$) along with constant source term indicating constant uplift rate. The linear stability analysis done in [1] reveals that small perturbation around the unchannelized analytical solution for LEM grows as $C_{\mathcal{I}_-} \approx 37$ in a semi-infinite domain with first channels appearing with periodic valley spacing.

This indicates that the unchannelized solution is always present, but becomes unstable to small perturbation (to any perturbation not only the noise in the initial condition, which is just one way that we have used to achieve that) as the critical value of channelization index is reached and that perturbation will grow and cause the channelization above the threshold value of parameters. To perform that small perturbation in the discretized model, we use a small amount of noise in the initial condition for a rectangular domain with a high aspect ratio. In the numerical implementation of the LEM that employs the same algorithm and same way of a small amount of spatial noise in initial condition as used in this work [2], it is shown that $C_{\mathcal{I}} \approx 35$ is the point of the first channelization using numerical experiment, which matches with the result of linear stability analysis. The agreement of the numerical model's first channelization with the linear stability analysis results in the original model shows that the role of the initial condition's small noise is to merely cause the numerical perturbation around the unstable analytical solution, and it does not affect the overall solution.

We agree with the reviewer that this should be clarified in the manuscript and hence we have added it in the updated text (Lines 225-228):

Numerical experiments are started from a linear initial condition containing a small random spatial noise. The added random noise produces small numerical perturbations around the smooth analytical solution (Equation (3.2)) so that, when the latter is unstable, channel instabilities grow resulting in the formation of spatial patterns.

We believe that the asymmetry of the solution is mainly caused by the mismatch of the solution geometry with the finite domain geometry used for simulations. The effect of domain shape on the non-symmetric spacing of valley is discussed in [1]. Figure 5 in [1] presents the first channelization for varying shape of rectangular domain with fewer defects for aspect ratio (length to width) 4.6 compared to 5.1. The same types of defects in the spacing of periodic straight channels (near the first channelization) can be observed in Figure 4(c,d) of the presented model. This indicates that these defects and the related symmetry breaking initiate primarily due to the mismatch between solution geometry with geometry taken in the model.

We modified the text to reflect this comment as follows in the revised manuscript (Lines 259-274):

The coupled supply and drainage networks obtained as steady-state solutions near the first channelization have a non-uniform gap between them, as shown in Figure 4. These symmetry-breaking irregularities, or dislocation defects, in the channel spacing are essentially created by the mismatch between the solution geometry with the domain geometry used

in the model, a behavior which is not unexpected in nonlinear pattern-forming systems [22, 23]; see [1] for a discussion of the effect of domain shape on the non-symmetric spacing in the drainage network near the first channelization.

We also verify that the spatial randomness added to the linear initial condition to trigger the channel instabilities does not alter the overall solution creating asymmetric valley spacing. Adding random samples from a uniform distribution over $[0, \hat{u}_r]$ for three levels of \hat{u}_r ($10^{-2}, 10^{-3}, 10^{-4}$), we observe the steady-state solutions for the same rectangular domain and fixed value of $\mathcal{C}_{\mathcal{I}\pm} = 10$ (near the first channelization). As presented in Figure R5(a), the steady-state scalar field \hat{h} distributions agree very well for three initial conditions. The obtained coupled networks of supply and drainage networks for $\hat{u}_r = 10^{-2}$ and $\hat{u}_r = 10^{-3}$ are also shown in Figure R5(b,c). For both cases, as expected, we obtain an equivalent distribution of both material densities with primary channels starting from boundaries having small defects in the spacing.

Figure R5: Steady-state simulation results for $\mathcal{C}_{\mathcal{I}\pm} = 10$ using three different perturbations in the initial conditions for a rectangular domain (width = 100, length = 500) with grid spacing $\Delta x = \Delta y = 1$. (a): Distribution of the scalar field, h , for $\hat{u}_r = 10^{-2}$ (black), $\hat{u}_r = 10^{-3}$ (red) and $\hat{u}_r = 10^{-4}$ (green). (b,c): Simulation results for the accumulation of (red) a_+ and (blue) a_- for $\hat{u}_r = 10^{-2}$ and $\hat{u}_r = 10^{-3}$, respectively. The white curve represents the interface $a_+ = a_-$.

The role of the random spatial noise used in the initial conditions for numerical perturbation is likely secondary, and occurs for very high values of channelization index. For a high value of $\mathcal{C}_{\mathcal{I}\pm}$ in the presented model, the feedback mechanism shows strong tendency of both materials to form channels. In the case of either disproportionate or comparable values of $\mathcal{C}_{\mathcal{I}\pm}$, we get heavy tendency of primary channel instability to coalesce and grow the networks. Now, in a discretized rectangular domain with finite resolution and each discrete point having 8 neighbours, these networks can be resolved up to finite grid spacing level and the flow-network also gets computed up to that resolution only. Therefore, during branching in heavy channelization regime (either branched or congested), mismatch of solution and domain geometry along with the finite resolution of numerical grid approximating flow-network and the discretization error lead to defects and symmetry breaking in the formed branching structures.

Accordingly, we have added a new paragraph discussing this in the revised manuscript (Lines 430-440):

Initial and boundary conditions, as well as the domain geometry, are crucial in the proposed model. In particular, dislocation defects in the spatial patterns are expected to arise whenever the solution geometry does not match the domain geometry, as typical of pattern-forming systems [22, 23]. These are evident in Figure 4(c,d), where the first-order networks form with small defects in the channel spacing. Increasing values of $\mathcal{C}_{\mathcal{I}\pm}$ drive the formation of complex networks, which are resolved up to the grid spacing. The reduced accuracy in the network approximation for heavy channelization is reflected on the imprint of the initial condition on the final steady-state solution for a very value of high channelization index, as the obtained channels get stuck due to the finite resolution of the discretized domain. Future work will be devoted to analyze the combined effect of these factors (geometry shape, grid resolution and alignment, initial and boundary conditions) on the uniqueness of numerical steady-state solutions in the proposed model.

I guess that the dynamics of the proposed model is highly dependent upon initial conditions. If this is true, then the random component is fundamental to achieve the shown results, which are not as universal as the authors state. I am not saying that this observation mines the validity of the work. What I am saying is that the authors should discuss this and be transparent about it. There are a number of published models that have this behavior. It should be acknowledged and explored.

This is related to the above point of small perturbation around unstable analytical solution and hence, as suggested by the reviewer, we now mention it in the starting of Section (4) that small amount of perturbation is required to get the branching patterns in the model (Lines 225-228):

Numerical experiments are started from a linear initial condition containing a small random spatial noise. The added random noise produces small numerical perturbations around the smooth analytical solution (Equation (3.2)) so that, when the latter is unstable, channel instabilities grow resulting in the formation of spatial patterns.

To further clarify the point that initial conditions do not drastically change the obtained solutions, and the smooth analytical solution is unstable to any small perturbation (not only the small spatial noise), we present here 10 simulations performed for the original Landscape Evolution Model (with only erosion, constant uplift as the source term ($\mathcal{C}_{\mathcal{I}+}$)) using the same numerical algorithm [2]. We use a square domain and fixed zero elevation value boundary conditions with two different initial conditions to verify that the obtained patterns are not highly dependent on the initial conditions. For the first case (Figure R6(a)), we take a small amount of random noise (same as applied to the linear condition in the presented model for coupled networks), and for the second case (Figure R6(b)), we employ a tiny sinusoidal elevated region with no random noise in the remaining parts of the domain to cause small numerical perturbations. We analyze five cases for different channelization indices ranging from 20 to 500 for both initial conditions.

As Figure R6 indicates, the obtained numerical solutions agree well for both cases with small dislocation defects occurring as $\mathcal{C}_{\mathcal{I}}$ is increased in both cases. The distribution of a (same as a_- in the presented model) for 10 cases further clarifies that random perturbation in initial conditions do not change the type of solution that is obtained. For

Figure R6: Simulations results for the original LEM in a square domain with side 100 m long and grid spacing of 1 m. (a,b): Initial conditions used in simulations, random noise and sinusoidal wave respectively. (c-g): The resulting a (same as a_- of the presented model) field indicating obtained ridge/valley patterns for varying values of C_I and the initial condition (a). (h-l): The resulting a field indicating obtained ridge/valley patterns for varying values of C_I and the initial condition (b). (j-n): Distribution of a with given C_I for initial condition (a) as shown in blue and initial condition (b) as shown in green curve.

$C_I = 20$, we get no channel in the domain and the resulting a field for both initial conditions match completely. As we increase the value C_I further, first channel occurs around $C_I \approx 40$ in the domain with good agreement in the distribution of a for both initial conditions. Further pushing the value of C_I results in secondary and tertiary branching in the patterns. A small difference in the a field for very high value of $C_I = 500$, as shown in Figure R6 (g,l), can be attributed to the fact that solution geometry does not agree with domain geometry as well as the finite resolution of the numerical grid resolves the networks till the grid spacing only, which reduces the accuracy of approximation of the network as channelization index increases and we get more and more branching in the domain. This way, starting with an initial condition has a small bearing in the final steady-state (as they appear to reach a local minima for a given finite resolution) for a very high channelization index value. But the distribution of a for both cases (Figure R6 (n)) reveal

that the overall solution remains mostly unaltered even in this case.

- p9, 147-56.

A longer description of the numerical approach would make the paper more self contained. I had to look at the paper [41] to understand how this is done. The sentence that the method is "inspired from the notion of a flow network..." is hard to decipher. From reading in [41] I came up with the following intuition (is it correct?): a 5-point FD stencil is used for the discretization of the Laplacian and a sort-of upwind scheme is used for the gradient. The overall solution strategy seems a sort-of gradient descent algorithm where a time-splitting approach is used to accommodate diffusion. In any case, the solution strategy reminds me of the papers on the MAST approach proposed by Arico' and Tucciarelli some time ago. (C Arico, T Tucciarelli, MAST solution of advection problems in irrotational flow fields, *Advances in water resources* 30 (3), 665-685).

We have added a new paragraph in Section (4) to explain the overall numerical algorithm and the general category of "topological sorting" algorithms to which this algorithm belongs (Lines 231-246):

We utilize the efficient numerical algorithm presented in [2] to update the scalar field h over the entire domain until the steady-state is reached. The numerical scheme consists of a time-splitting approach where we use a five-point stencil second-order central-difference formula for discretizing the Laplace operator for the diffusion and a modified breadth-first topological sorting algorithm for implicitly updating the nonlinear source/sink term. The sorting algorithm presented in [2] belongs to the so-called 'task-scheduling' problems, where the edges of the network symbolize the tasks' dependency. In this model, edges represent the relationship amongst points in the flow-distribution network of the material, which is traversed in a way to make the matrix system upper/lower triangular for the efficient implicit computation. The development and the applications of this category of sorting algorithm are discussed in [24, 25, 26, 27, 28].

The accuracy of the employed numerical algorithm has been carefully tested for the case of the drainage-network evolution model in the natural landscape against analytical solutions in non-channelized/vascularized conditions as well as against analytical results of linear stability analysis [1, 2]. The temporal evolution of the mean-field solution in the fully channelized/vascularized regime agrees with the exact analytical expression for the transient solution [2]. We refer to these references for further details.

- p11, fig5.

I do not understand the lack of symmetry. I would expect only straight lines for equal values of C_I . In the other cases, I would expect a recurring (periodic) pattern of branching structures. Why do these structures emerge only in the right portion of the domain? If this is the effect of the random component of the initial conditions, it should be thoroughly discussed, as it could be a symptom of instability of the numerical scheme or the model.

In addition, the gradient descent algorithm could be trapped in some local minima away from the minimizer. Overall, it is a diffusion process and should maintain some sort of symmetry. Please discuss.

We have modified the manuscript to include this discussion. In this minimalist model of coupled networks as well as the original landscape evolution model, the solution begins with initial period of channel formation instability from the boundaries, with primary channels coalescing together when a certain threshold of $\mathcal{C}_{\mathcal{I}_{\pm}}$ is reached to create secondary branching. Starting above a certain value of $\mathcal{C}_{\mathcal{I}_{\pm}}$ near first channelization, we get straight primary channels with already small defect in the ridge/valley spacing as the solution geometry does not exactly match with finite domain geometry used for simulations (Figure 4 (c,d) of the manuscript).

These defects are also shown to occur in landscape evolution models, referred as dislocation defects, as shown in Figure 5 of [1]. The varying shape of domain geometry in the landscape evolution model affect these defects in the solution with less defect (non-symmetric valley spacing) occurring for the rectangular domain with aspect ratio 4.6 compared to the domain with aspect ratio 5.1.

As the values of $\mathcal{C}_{\mathcal{I}_{\pm}}$ are pushed beyond this range to higher values (relatively low value of diffusion), these primary channel instabilities grow, and in some cases coalesce breaking the symmetry of the solution. In addition, the numerical solution may get trapped in local minima for a very high value of channelization index due to discretization errors, finite resolution of the grid, as well as alignment of flow-distribution network with the computational grid. As a result, initial conditions may influence the final result for a very high value of channelization index [1, 29].

As we mentioned before in the response to the related remark by the reviewer, this is discussed clearly in the revised manuscript (Lines 430-440):

Initial and boundary conditions, as well as the domain geometry, are crucial in the proposed model. In particular, dislocation defects in the spatial patterns are expected to arise whenever the solution geometry does not match the domain geometry, as typical of pattern-forming systems [22, 23]. These are evident in Figure 4(c,d), where the first-order networks form with small defects in the channel spacing. Increasing values of $\mathcal{C}_{\mathcal{I}_{\pm}}$ drive the formation of complex networks, which are resolved up to the grid spacing. The reduced accuracy in the network approximation for heavy channelization is reflected on the imprint of the initial condition on the final steady-state solution for a very value of high channelization index, as the obtained channels get stuck due to the finite resolution of the discretized domain. Future work will be devoted to analyze the combined effect of these factors (geometry shape, grid resolution and alignment, initial and boundary conditions) on the uniqueness of numerical steady-state solutions in the proposed model.

- p14, l45-52. I am repeating and reinforcing here the point made in p11, fig5. I assume that also here the initial conditions are randomly perturbed. Again I am concerned by the fact that the lack of symmetry and the emergence of the preferential flow paths could be an undiscussed consequence of the random perturbation of the

initial conditions. I'm always worried when these singular structures emerge as a consequence of randomness. It means that instabilities are present and not under control. Do you get the same results when no randomness is added to the initial conditions? If yes, then the final equilibrium is a stable equilibrium, if not, then the reader should be warned that the steady state equilibrium point is not unique (another equivalent way to put it is that the gradient descent algorithm is stopping in a local minimum which may be close to the global minimum but far from the global minimizer). In any case, in these situations numerical errors could be the triggers for the branching patterns. If this is the case, it should be discussed. In my view this is not necessarily detrimental to the manuscript. There are many models that behave like this as e.g. in your reference [12]. But in this situation the results should be analyzed using different strategies. I completely understand that this is beyond the scope of the current paper, but it should be mentioned to give a complete picture to the reader.

This point is related to the previous comment. We agree with the reviewer that the initial condition with small random noise triggers spatial patterns in the domain by perturbing the unstable analytical solution. However, as explained previously, any perturbation around the unstable smooth solution will trigger these patterns above a threshold value of parameters (adding small amount of noise is just one of the way). In the revised version, we clarify this point at the start of Section (4) (Lines 225-228):

Numerical experiments are started from a linear initial condition containing a small random spatial noise. The added random noise produces small numerical perturbations around the smooth analytical solution (Equation (3.2)) so that, when the latter is unstable, channel instabilities grow resulting in the formation of spatial patterns.

The added perturbations to the linear initial condition do not change the obtained solutions. We clarified it in the previous response by showing numerical simulation results to be in good agreement for different levels of random noise in the initial conditions. We show it for the case $C_{I\pm} = 10$ in a rectangular domain (width = 100, length = 500), where we start with different perturbation around the linear initial condition. The obtained solutions are observed to be in good-agreement with each other.

As we mention in the previous response on the role of the initial conditions in the model, this numerical experiment has been added as a part of the Code verification section (Sec 4(a)(i)) in lines 259-274 of the revised manuscript:

The coupled supply and drainage networks obtained as steady-state solutions near the first channelization have a non-uniform gap between them, as shown in Figure 4. These symmetry-breaking irregularities, or dislocation defects, in the channel spacing are essentially created by the mismatch between the solution geometry with the domain geometry used in the model, a behavior which is not unexpected in nonlinear pattern-forming systems [22, 23]; see [1] for a discussion of the effect of domain shape on the non-symmetric spacing in the drainage network near the first channelization.

We also verify that the spatial randomness added to the linear initial condition to trigger the channel instabilities does not alter the overall solution creating asymmetric valley spacing. Adding random samples from a uniform distribution over $[0, \hat{u}_r)$ for three levels of \hat{u}_r (10^{-2} , 10^{-3} , 10^{-4}), we observe the steady-state solutions for the same rectangular

domain and fixed value of $\mathcal{C}_{\mathcal{I}_{\pm}} = 10$ (near the first channelization). As presented in Figure R7(a), the steady-state scalar field \hat{h} distributions agree very well for three initial conditions. The obtained coupled networks of supply and drainage networks for $\hat{u}_r = 10^{-2}$ and $\hat{u}_r = 10^{-3}$ are also shown in Figure R7(b,c). For both cases, as expected, we obtain an equivalent distribution of both material densities with primary channels starting from boundaries having small defects in the spacing.

Figure R7: Steady-state simulation results for $\mathcal{C}_{\mathcal{I}_{\pm}} = 10$ using three different perturbations in the initial conditions for a rectangular domain (width = 100, length = 500) with grid spacing $\Delta x = \Delta y = 1$. (a): Distribution of the scalar field, h , for $\hat{u}_r = 10^{-2}$ (black), $\hat{u}_r = 10^{-3}$ (red) and $\hat{u}_r = 10^{-4}$ (green). (b,c): Simulation results for the accumulation of (red) a_+ and (blue) a_- for $\hat{u}_r = 10^{-2}$ and $\hat{u}_r = 10^{-3}$, respectively. The white curve represents the interface $a_+ = a_-$.

We have also modified the text to explain the role of domain geometry and discretization in resolving the obtained networks for a high value of $\mathcal{C}_{\mathcal{I}_{\pm}}$. For heavy channelization, symmetry breaking occurs due to the defect of solution geometry and domain geometry mismatch, discretization error, the approximation for the flow-distribution network to be aligned with underlying finite resolution grid occur, for which the solution gets trapped in a local minimum. Therefore, the local minimum (apparent steady-state of the numerical model) has the bearing of the randomly perturbed initial conditions for high value of channelization indices.

We have added a new paragraph in the Conclusion section of the revised manuscript (presented in the previous response) to give a complete picture to the reader (Lines 430-440).

MINOR REMARKS:

- **p4, l20-24. Here the authors introduce a scalar field but it is not clear what this scalar field is. Later on the authors assume that the transport velocity is proportional to the gradient of this scalar field and one may regard it as a potential field, and this could be anticipated here for better clarity.**

Thank you for pointing this out. We have added this point in Section 2(a) (Lines 62-65):

This way, the scalar field, h , can be perceived as a potential field guiding the material flow, which results in the distribution of material density, say a_- , as shown in Figure 1b, highlighting the drainage network for the topography.

- **p4, l43. Awkward sentence. May be change "enter" in "entering"**

We have modified the sentence in the updated manuscript (Lines 85-87):

One can envision the supplied material density a_+ entering the area at the boundary concentrated at the ridges of the scalar field h , flowing and getting distributed over the hillslopes as it gets exhausted.

- **p5, eq. (2.2). The authors say "for simplicity" we assume... I am puzzled by this "simplicity" assumption. Simple as opposed to what? This is not simple at all and it is possibly this choice that sparks the richness of the solutions of the proposed model. I would like to see a few comments on this.**

Thanks for the remark. By simplicity we meant that both the materials are assumed to follow the gradient of field h with unit speed in this model. There can be several variations on top of of this assumption in the literature for network evolution models [10, 12]. For example - diffusion may take place at every location, the speed may vary based on the slope instead of being constant, location of the domain, etc. These all changes can be done for either or both materials, which will affect the obtained spatial patterns of networks. Since we are interested in this concept of multiple materials flowing and carving their networks in the same domain and the whole feedback loop in a minimalist way, and not looking at the role of change in the particular mathematical form of the movement laws, we keep the analysis simply to the assumption of unit speed along the gradient of h .

We have added a paragraph about this in the Conclusions section of the updated manuscript (Lines 422-427):

In the formulation of this minimalist model, we assume that the material flows along the direction of the steepest descent of field h with unit speed (Equation (2.2)). There are network evolution models in geomorphology, hydrology, chemotaxis and vasculogenesis, where the flow has different spatiotemporal scales [8, 10, 12]. Different laws of material transport and diffusion properties can be employed in the presented numerical scheme for any loop-less flow-distribution network.

- p6, l7. Why the adjective "nonlocal"? Does it bear some significance in addition to "distributed"? If yes it should be explained, if not, then it should be avoided as it resembles other types of processes.

We agree with the reviewer. The word "nonlocal" should be omitted from the manuscript to avoid any confusion. We have updated lines 99-101 as:

The scalar field h is assumed to co-evolve with the density fields of both materials. Specifically, the temporal evolution of h consists of a diffusion term and nonlinear sink and source terms due to the feedback from both materials as

- p7, l42-50. Also here this discussion about bcs is confusing and does not add a lot of insights. Only conditions on top and bottom boundaries are discussed and set to homogeneous Dirichlet for the densities. If no flow is set on the other two boundaries then again there is only internal redistribution of densities, is it right?

As said before, we agree with the reviewer that the top and bottom boundaries are set to homogeneous Dirichlet for the densities with closed side boundary conditions for the internal redistribution of densities. This is clearly written in the new Section 2(c) on Boundary conditions for the presented model.

- p7, eq. (2.7). This procedure is valid only if the production rates r are constant in space. Should be mentioned.

We agree with the reviewer that the simplification to 2-field model is valid for a constant value of r_{\pm} and mention it in the revised manuscript (Lines 177-178):

It is interesting to observe that the model can be reduced to a 2-field system for the case $m_{\pm} = n_{\pm} = 1$ and constant r_{\pm} over the spatial domain.

- p10, l8-16. Need more info on the simulation. For example, what is the grid size that allows the resolution of the ridges/valleys displayed in the figures? Does the solution change as the mesh is refined? I believe that grid-alignment problems should emerge. Has this been noticed?

We now clearly mention in the revised manuscript that the simulations are performed with unit grid spacing for all the presented results. We write it in the starting of Section (4) Numerical solutions (Lines 229-230):

Raster grid with a spacing of one unit is used in the numerical simulations for 2-/3-dimensional cases.

We also mention it in the caption of various figures of the updated manuscript.

Caption of Figure 4: (a): Solid lines represent the computed mean surface profile (\bar{h}) along the length in a rectangular domain (width = 100, length = 500) with grid spacing $\Delta x = \Delta y = 1$ for $C_{\mathcal{I}_{\pm}} = 1, 3.5$ and 12.5 ...

Caption of Figure 5: Steady-state simulation results for $C_{\mathcal{I}_{\pm}} = 10$ using three different perturbations in the initial conditions for a rectangular domain (width = 100, length = 500) with grid spacing $\Delta x = \Delta y = 1$.

Caption of Figure 6: Simulation results for various values of $C_{\mathcal{I}_+}$ and $C_{\mathcal{I}_-}$ in a rectangular domain (width = 100, length = 500) with grid spacing $\Delta x = \Delta y = 1$.

Caption of Figure 10: Dependence of critical $C_{\mathcal{I}_{\pm}}$ in a rectangular domain with high aspect ratio (width = 100, length = 500) with grid spacing $\Delta x = \Delta y = 1$.

Caption of Figure 12: Simulation results for the 3-dimensional domain ($x = 50, y = 80, z = 60$) with grid spacing $\Delta x = \Delta y = \Delta z = 1$.

We agree with the reviewer that as the grid spacing increases, the issue of flow-alignment with grid increases the numerical error. This is expected as for the rectangular grid used in this work, has 8 neighbors for the flow-distribution at every point. We have not noticed this issue at the unit grid spacing used in the simulations.

Figure R8: Steady-state solutions for a square domain (side length = 20 m), $m = 0.5, n = 1.0, D = 5.0 \times 10^{-3} \text{ m}^2 \text{ year}^{-1}, U = 5.0 \times 10^{-5} \text{ m year}^{-1}$, and $C_{\mathcal{I}} = 62$. (a), (b) and (c) represent the ridge/valley network for $\Delta x = 0.25, \Delta x = 0.125$ and $\Delta x = 0.0625$ respectively (brown = ridge, green = valley). (d): Normalized hypsometric curves for $\Delta x = 0.5, \Delta x = 0.25, \Delta x = 0.125$ and $\Delta x = 0.0625$. (e): Linear relationship between pseudo error versus grid spacing with slope equal to 0.98 indicates the first-order spatial accuracy of the implemented numerical algorithm.

The obtained spatial patterns, though less accurate on increasing spacing of the underlying mesh, are grid-independent in nature. We discuss it here using an example of the original numerical model from the landscape evolution equation in [2] (Section 5.1.2), where the sink term of this model is employed in the governing equations with the constant uplift rate (for a real natural landscape with fixed zero elevation boundary along the domain side and $C_{\mathcal{I}_+} = 0$).

In the numerical experiment, we took four cases of a square domain with side 20 m and fixed values of parameters with grid resolution $\Delta x = 0.5$, $\Delta x = 0.25$, $\Delta x = 0.125$ and $\Delta x = 0.0625$ (Figure R8). We showed that the solution is grid-independent for all cases, as only one first-order channel along each domain boundary is obtained in each case, and the distribution of h (which represents the height of real landscape in this model) also remains unaltered (Figure R8 (d)). The accuracy of the obtained numerical results goes down as first-order when the grid spacing of the raster mesh is expanded, which is expected.

- p10, l44-49. What are the BC's for a_+ and a_- ? I assume one and zero, but please specify. In a simpler way, you change the values of the C_I by changing only the consumption rates, right?

The reviewer is correct in pointing out that the value of $C_{\mathcal{I}_{\pm}}$ are modified by changing the consumption/production rate. The boundary conditions are same as discussed in Sec 2(c) of the updated manuscript (closed BC on sides for both a_{\pm} , with $a_+ = 0$ at bottom edge, and $a_- = 0$ at top edge), and the same are used in the numerical simulations.

We mention it explicitly for better clarity to reader in the starting of Section (4) (Lines 228-229):

The boundary conditions in all the numerical simulations for 2-/3-dimensional cases are taken the same as discussed in Section 2(c).

Awkward sentence at line 47: "Having all other parts parameters..."

We have modified this sentence (Lines 279-281):

Keeping the value of all other parameters fixed, the change in these rates can be expressed as the modification in two non-dimensional channelization indices, $C_{\mathcal{I}_+}$ and $C_{\mathcal{I}_-}$.

- p11, l52. "as opposed two-field model" should be "as opposed to the two-field model"

We have corrected this part (Lines 304-306):

This shows that the non-trivial boundary conditions allowed by the three-field model (as opposed to the two-field model) influence the solution throughout the domain, both quantitatively and qualitatively. A full discussion of this is included in Appendix 6.

- p11, l56. "hike" should be may be "hide"

We have modified the sentence (Lines 308-310):

Increasing the value of the respective channelization index enhances the feedback of supplied (a_+) and drained material (a_-) to actively form ridges and valleys of the 2-dimensional 'landscape'.

- p11, can not -> cannot

We have changed this sentence in the revised manuscript (Lines 315-317):

This tendency of both networks to grow, competing against each other, makes them stuck in a large number of primary channels.

- p13, l7. "less number" -> "smaller number"

We have updated this part (Lines 332-333):

More branching results in smaller number of main channels for $C_{\mathcal{I}+} \ll C_{\mathcal{I}-}$, as indicated by the blue in Figure 9(B).

- p14, l7-11. The sentence is convoluted. Please, try to rephrase to make it of more immediate understanding.

Thank you for the suggestion. We have restructured this part of the paragraph to have a clear understanding for the reader (Lines 382-391):

The steady-state solutions for the 3-dimensional case agree with the patterns observed in the 2-dimensional results. As shown in Figure 12(a), a large number of red contour curves (high-density region of h) in cross-section near the face $h(x, 0, z)$ extend over the domain and reach the cross-section near the face $h(x, 80, z)$, which is dominated by the blue contour curves (low-density region of h). This spatial pattern for $C_{\mathcal{I}\pm} = 1000$ is similar to the solutions obtained in the 2-dimensional case for the comparable value of two channelization indices, where a large number of shallow ridges and valleys, starting from either edge of the rectangular domain, propagate to the opposite end. We display the largest drainage conduit from the steady-state solution in Figure 12(c,d), where green haze in panel c indicates the points in the domain from which the flow is collected in the given conduit.

References

- [1] Sara Bonetti, Milad Hooshyar, Carlo Camporeale, and Amilcare Porporato. Channelization cascade in landscape evolution. *Proceedings of the National Academy of Sciences*, 2020.
- [2] Shashank Kumar Anand, Milad Hooshyar, and Amilcare Porporato. Linear layout of multiple flow-direction networks for landscape-evolution simulations. *Environmental Modelling & Software*, 133:104804, 2020.

- [3] Tom J Coulthard. Landscape evolution models: a software review. *Hydrological processes*, 15(1):165–173, 2001.
- [4] Björn Birnir, Terence R Smith, and George E Merchant. The scaling of fluvial landscapes. *Computers & geosciences*, 27(10):1189–1216, 2001.
- [5] Erkan Istanbuluoglu and Rafael L Bras. Vegetation-modulated landscape evolution: Effects of vegetation on landscape processes, drainage density, and topography. *Journal of Geophysical Research: Earth Surface*, 110(F2), 2005.
- [6] J Taylor Perron, Paul W Richardson, Ken L Ferrier, and Mathieu Lapôtre. The root of branching river networks. *Nature*, 492(7427):100, 2012.
- [7] Andrew Fowler. *Mathematical geoscience*, volume 36. Springer Science & Business Media, 2011.
- [8] Thomas Hillen and Kevin J Painter. A user’s guide to pde models for chemotaxis. *Journal of mathematical biology*, 58(1-2):183, 2009.
- [9] Evelyn F Keller and Lee A Segel. Initiation of slime mold aggregation viewed as an instability. *Journal of theoretical biology*, 26(3):399–415, 1970.
- [10] D Manoussaki, SR Lubkin, RB Vemon, and JD Murray. A mechanical model for the formation of vascular networks in vitro. *Acta biotheoretica*, 44(3-4):271–282, 1996.
- [11] Daphne Manoussaki. A mechanochemical model of angiogenesis and vasculogenesis. *ESAIM: Mathematical Modelling and Numerical Analysis*, 37(4):581–599, 2003.
- [12] Chengzhi Qin, A-X Zhu, Tao Pei, Baoluo Li, Chenghu Zhou, and L Yang. An adaptive approach to selecting a flow-partition exponent for a multiple-flow-direction algorithm. *International Journal of Geographical Information Science*, 21(4):443–458, 2007.
- [13] Alex Chen, Jérôme Darbon, Giuseppe Buttazzo, Filippo Santambrogio, and Jean-Michel Morel. On the equations of landscape formation. *Interfaces and Free Boundaries*, 16(1):105–136, 2014.
- [14] Mark AJ Chaplain. Mathematical modelling of angiogenesis. *Journal of neuro-oncology*, 50(1-2):37–51, 2000.
- [15] Jan Haskovec, Peter Markowich, Benoît Perthame, and Matthias Schlottbom. Notes on a pde system for biological network formation. *Nonlinear Analysis*, 138:127–155, 2016.
- [16] A-L Barabási and Harry Eugene Stanley. *Fractal concepts in surface growth*. Cambridge university press, 1995.
- [17] Joachim Krug. Origins of scale invariance in growth processes. *Advances in Physics*, 46(2):139–282, 1997.
- [18] Mehran Kardar, Giorgio Parisi, and Yi-Cheng Zhang. Dynamic scaling of growing interfaces. *Physical Review Letters*, 56(9):889, 1986.
- [19] Jon D Pelletier. Fractal behavior in space and time in a simplified model of fluvial landform evolution. *Geomorphology*, 91(3-4):291–301, 2007.
- [20] Thomas Hillen, Kevin Painter, and Christian Schmeiser. Global existence for chemotaxis with finite sampling radius. *Discrete & Continuous Dynamical Systems-B*, 7(1):125, 2007.

- [21] Mark Alber, Nan Chen, Tilmann Glimm, and Pavel M Lushnikov. Multiscale dynamics of biological cells with chemotactic interactions: from a discrete stochastic model to a continuous description. *Physical Review E*, 73(5):051901, 2006.
- [22] Mark C Cross and Pierre C Hohenberg. Pattern formation outside of equilibrium. *Reviews of modern physics*, 65(3):851, 1993.
- [23] Pablo G Debenedetti and Frank H Stillinger. Supercooled liquids and the glass transition. *Nature*, 410(6825):259–267, 2001.
- [24] Jun Ma, Kazuo Iwama, Tadao Takaoka, and Qian-Ping Gu. Efficient parallel and distributed topological sort algorithms. In *Proceedings of IEEE International Symposium on Parallel Algorithms Architecture Synthesis*, pages 378–383. IEEE, 1997.
- [25] Costanza Arico and Tullio Tucciarelli. Mast solution of advection problems in irrotational flow fields. *Advances in water resources*, 30(3):665–685, 2007.
- [26] David J Pearce and Paul HJ Kelly. A dynamic topological sort algorithm for directed acyclic graphs. *Journal of Experimental Algorithmics (JEA)*, 11:1–7, 2007.
- [27] Jean Braun and Sean D Willett. A very efficient $O(n)$, implicit and parallel method to solve the stream power equation governing fluvial incision and landscape evolution. *Geomorphology*, 180:170–179, 2013.
- [28] Richard Barnes. Accelerating a fluvial incision and landscape evolution model with parallelism. *Geomorphology*, 330:28–39, 2019.
- [29] J Taylor Perron and Sergio Fagherazzi. The legacy of initial conditions in landscape evolution. *Earth Surface Processes and Landforms*, 37(1):52–63, 2012.